# Cytogenetic signatures favoring metastatic organotropism in colorectal cancer

Mariola Monika Golas [1,2] ✉, Bastian Gunawan[3,9], Angelika Gutenberg[4], Bernhard C. Danner[5], Jan S. Gerdes [3,10], Christine Stadelmann [6], Laszlo Füzesi[3], Torsten Liersch[7] & Bjoern Sander [8] ✉

Colorectal carcinoma (CRC) exhibits metastatic organotropism, primarily targeting liver, lung, and rarely the brain. Here, we study chromosomal imbalances (CIs) in cohorts of primary CRCs and metastases. Brain metastases show the highest burden of CIs, including aneuploidies and focal CIs, with enrichment of +12p encoding *KRAS*. Compared to liver and lung metastases, brain metastases present with increased co-occurrence of *KRAS* mutation and amplification. CRCs with concurrent *KRAS* mutation and amplification display significant metabolic reprogramming with upregulation of glycolysis, along-side upregulation of cell cycle pathways, including copy number gains of *MDM2* and *CDK4*. Evolutionary modeling suggests early acquisition of many organotropic CIs enriched in both liver and brain metastases, while brain-enriched CIs preferentially emerge later. Collectively, this study supports a model where cytogenetic events in CRCs favor site-specific metastatic colonization. These site-enriched CI patterns may serve as biomarkers for metastatic potential in precision oncology.

Colorectal cancer (CRC) is responsible for about 10% of cancer-related deaths worldwide, which corresponds to ~900,000 deaths annually[1]. The vast majority of cancer-associated deaths are related to metastatic spread of the primary malignancy[2]. About one in four to five patients with CRC presents with distant metastasis at initial cancer staging[3], and about one in five patients diagnosed with CRC develops metachronous metastatic disease in the clinical course[4]. Metastatic CRC spread can occur very early in the progression of disease, at a time-point when the primary tumor is not yet necessarily clinically manifest[5–7]. In general, metastatic seeding can follow a polyclonal pattern, in which metastatic cells spread from multiple tumor clones, but it can also follow a monoclonal pattern[7–9]. Previous work indicates that in about two-thirds of CRC cases, lymph node and distant metastases appear to be derived from independent subclones[10]. In the course of CRC disease, the main site for metastasis is the liver, followed by the lung[11]. Less commonly, peritoneal spread is seen in CRC, and rare sites of metastasis include other gastrointestinal organs, bones, and the brain/central nervous system (CNS)[11]. Thus, the spread of CRC cells follows a specific organ pattern, a phenomenon referred to as metastatic organotropism[12,13].

Accumulating evidence supports a model in which successful colonization of cancer cells at a secondary site does not result merely from passive mechanical constraints, but rather involves active shaping of the microenvironment at the secondary site to create a permissive niche that facilitates cancer cell survival and growth at distant sites[14–18]. Cancer cells appear to act on the prospective secondary site

[1]Human Genetics, Faculty of Medicine, University of Augsburg, Augsburg, Germany. [2]Comprehensive Cancer Center Augsburg, University Medical Center Augsburg, Augsburg, Germany. [3]Institute of Pathology, University Medical Center Göttingen, Göttingen, Germany. [4]Department of Neurosurgery, Asklepios Hospital Harburg, Hamburg, Germany. [5]Department of Cardiac, Thoracic and Vascular Surgery, University Medical Center Göttingen, Göttingen, Germany. [6]Department of Neuropathology, University Medical Center Göttingen, Göttingen, Germany. [7]Department of General, Visceral and Pediatric Surgery, University Medical Center Göttingen, Göttingen, Germany. [8]Institute of Pathology, Hannover Medical School, Hannover, Germany. [9]Present address: Institute of Pathology Northern Hesse, Kassel, Germany. [10]Present address: Epilepsy Center Hamburg, Evangelical Hospital Alsterdorf, Neurology and Epileptology, Hamburg, Germany. ✉e-mail: monika.golas@med.uni-augsburg.de; sander.bjoern.pat@mh-hannover.de

to induce a microenvironment that favors invasion and colonization of the secondary organ[14–18]. The formation of this pre-metastatic niche is hypothesized to be induced by factors and extracellular vesicles, including exosomes released from the primary cancer cells[15,19,20]. In addition, evidence is accumulating that hematopoietic stem cell-derived cells shape the metastatic niche. For example, cancer cells appear to use tenascin C-activated macrophages to induce formation of a vascular niche in the target organ[18]. However, the genetic processes involved in metastatic organotropism remain poorly understood.

Modeling the tumor evolution may identify genetic determinants that confer metastatic organotropism. The process of tumor evolution is driven by the acquisition of a diverse array of genetic alterations, ranging from single nucleotide variants and small insertions/deletions (indels), to focal chromosomal aberrations and structural variants such as translocations and inversions, as well as chromosomal arm aneuploidies and whole chromosomal aneuploidies[21,22]. In particular, primary CRCs commonly carry a set of defined DNA copy number variations (CNVs), including gains of 8q, 13q, and 20q, as well as losses of 18q[23–28]. These CNVs often result from whole chromosomal arm and whole chromosome aneuploidies[24,29]. In addition to CNVs, mutations in a number of genes have been described: collaborative next-generation sequencing (NGS) efforts[23] have led to the description of a temporal sequence of driver mutation acquisition in the adenomatous polyposis coli gene (*APC*), the tumor protein gene P53 (*TP53*), the Kirsten rat sarcoma virus oncogene (*KRAS*), and the SMAD family mothers against decapentaplegic homolog 4 gene (*SMAD4*), among others[30]. This sequence is consistent with previous studies[31–34] that culminated in the proposal of a sequential progression model, which has become known as the Vogelgram[35]. However, the specific sequence of genetic events may not be as critical for CRC development as the overall accumulation of genetic alterations over time[35].

Following the Vogelgram model, a number of additional models of tumor evolution in CRC have been proposed. These evolutionary models integrate the acquisition of genetic changes in the form of gradual processes or bursts, combined with different extents of clonal selection[36]. In the linear model, of which the Vogelgram is a prototype, genetic events are acquired sequentially. In this model, genetic changes with a growth advantage lead to preferential survival of the fittest clone(s)[35]. The branched evolution[37] and the neutral model[38] share the idea of a parallel, i.e., branched, expansion of clones, but differ in the extent of selective pressure. While the former model assumes some degree of selection[37], the latter model favors unrestricted co-expansion of clones in the absence of relevant selection pressure[38]. In the fourth model, the punctuated model[39], also known as the big bang model[40], massive genetic changes occur in bursts during cancer development, followed by the expansion of clones with selective advantage[39] or under neutral evolutionary conditions[40,41].

Cytogenetic aberrations in tumor cells, such as those arising from chromoplexy of several chromosomes[39] and chromothripsis of single chromosomes[42] as well as chromosomal CNVs[43] have been linked to the punctuated model. In other tumors, it has been suggested that the acquisition of CNVs followed a more gradual evolutionary path[44]. Thus, while we have a detailed idea of possible acquisition patterns of genetic alterations, an understanding of the chronology of metastatic capacity acquisition, and of the genetic alterations driving metastatic organotropism, is only beginning to emerge[28,45].

Primary CRC and its metastases share a majority of somatic mutations with few exclusive mutations private to the primary tumor or metastasis[46,47]. In a study analyzing primary CRCs and their metastases, the vast majority of which colonized the liver, CRC metastasis-enriched mutations were identified in genes linked to the extracellular matrix, PI3K signaling, and stellate cell activation[47]. Also, involvement of TGF-beta signaling has been suggested in liver metastasis[45], as has RAS signaling with *KRAS* mutations in lung and brain metastases[48,49].

However, whether these mutations represent organotropic genetic alterations or a general expression of the metastatic potential remained largely unknown. Recent studies have shown no significant differences between the tumor mutational burden (TMB) in primary CRC and its metastases[28]. Along these lines, *KRAS* mutations were the only organotropic mutations in lung and brain metastases of CRC that have been repeatedly reported[28,45], yet the complexity of metastatic behavior, especially in less-studied sites like the brain, remains to be explored in detail.

At the same time, previous CRC studies on CIs have focused on genomic changes in primary tumors and single metastatic sites, which facilitated the definition of common chromosomal aberrations in primary CRC and specific metastases[23–28,50–63]. CRC is an example of a cancer entity characterized by a complex pattern of chromosomal imbalances (CIs)[23–28,50–63], and it has been suggested that chromosomal aberrations are acquired early in disease development[9]. Since CIs typically alter the expression of a (large) group of genes simultaneously[64], CIs can potentially have additive or even synergistic effects and be further potentiated by co-occurrence of multiple CIs.

In this work, we examine the cytogenetic profile of CRCs and their metastatic spread to the liver, lung, and brain by combining independent cohorts. The first cohort, termed CRCTropism, employs a dedicated cytogenetic approach, while the second cohort, the Memorial Sloan Kettering (MSK) MetTropism cohort[28], is based on NGS data. In the CRCTropism cohort, the vast majority of patients, including all cases with brain metastases, underwent tumor sampling prior to receiving any targeted therapy. This approach enables the identification of genetic signatures that support intrinsic organotropy rather than reflecting potential consequences of targeted therapeutic interventions. We demonstrate that metastases are characterized by site-enriched patterns of cytogenetic events, which translate into cytogenetic signatures for secondary sites. Using a third cohort from The Cancer Genome Atlas (TCGA)[65,66], we show that concurrent *KRAS* mutation and amplification, enriched in brain metastases, is associated with enhanced metabolic reprogramming and typically coupled with *MDM2* and *CDK4* co-amplifications already in primary tumors. Organotropic patterns reveal the enrichment of specific CNVs in liver and brain metastases, their rarity in lung metastases, and a temporal dimension defined by the order in which these alterations are acquired during tumor evolution.

## Results

### Characterization of the CRCTropism cohort

The CRCTropism cohort comprised a total of 314 tumors derived from 191 patients diagnosed with metastatic CRC. Demographic and clinic-pathological data of this cohort are summarized in Supplementary Table 1, and an overview of the analyses is given in Supplementary Fig. 1. The cohort consists of 80 primary CRCs as well as 117, 78, and 39 metastases to the liver, lung, and brain, respectively. In some cases, multiple tumor tissue samples were analyzed, including samples from different areas of the primary tumor, recurrences, and/or one or more metastases. For 80 patients, at least one primary CRC/metastasis (P/M) pair was available, totaling 97 P/M pairs. The comparative analysis of the primary CRCs reported in this study and TCGA cohort[65,66], which exclusively comprises primary CRCs, revealed no statistically significant differences, except for the subgroup of primary CRCs associated with lung metastases. This subgroup exhibited a significantly lower prevalence of +7q (0.426 in the TCGA cohort vs. 0.143 in primary CRC with lung metastases [log2 ratio -1.57]; $p_{adjust} = 0.013$; Fisher test with Benjamini-Hochberg correction of multiple testing) and -18p (0.568 vs. 0.194 [log2 ratio -1.55]; $p_{adjust} = 4.17 \times 10^{-3}$) and +20p (0.402 vs. 0.074 [log2 ratio -2.44]; $p_{adjust} = 0.013$), concomitant with a higher frequency of -20p (0.068 vs. 0.296, [log2 ratio 2.12]; $p_{adjust} = 0.013$; Supplementary Fig. 2).

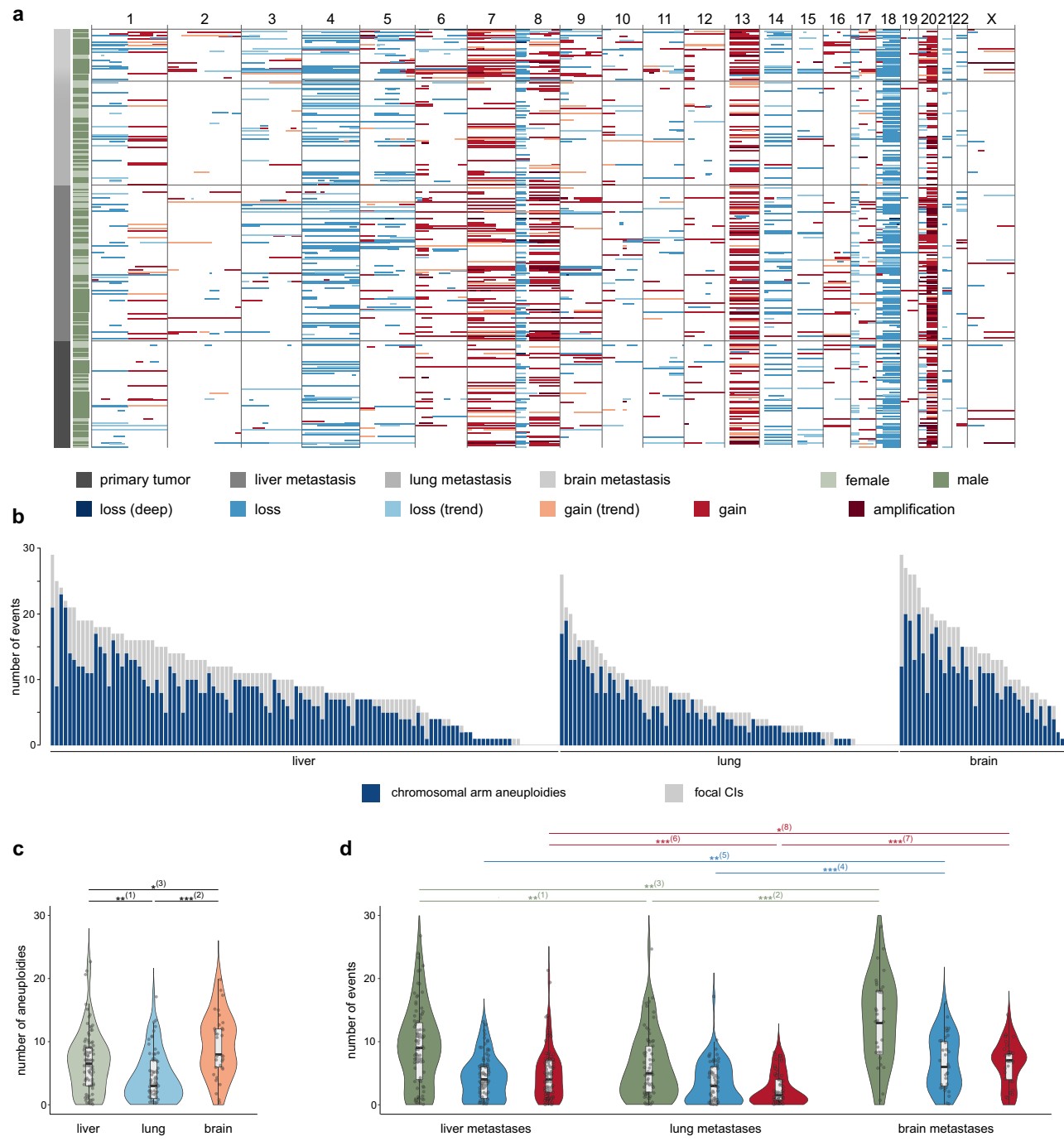

**Fig. 1 | Pattern of CIs observed in the CRCTropism cohort. a** Chromosomal losses and gains in shades of blue and red, respectively, observed in the cohort with 314 tumors, including $n = 80$ primary CRC and $n = 234$ distant CRC metastases in the liver, lung, and brain. On the left, tumor site and sex are color-encoded according to the legend. **b** Number of apparent chromosomal arm aneuploidies in relation to focal CIs, visualized as a stacked bar chart ($n = 234$). **c** Distribution of chromosomal arm aneuploidies in CRC metastases ($n = 191$). (1) $p_{adjust} = 2.96 \times 10^{-3}$; (2) $p_{adjust} = 6.94 \times 10^{-6}$; (3) $p_{adjust} = 1.31 \times 10^{-2}$. **d** Distribution of CIs including losses and gains for CRC metastases ($n = 191$). (1) $p_{adjust} = 2.46 \times 10^{-3}$; (2) $p_{adjust} = 1.37 \times 10^{-6}$;

(3) $p_{adjust} = 5.57 \times 10^{-3}$; (4) $p_{adjust} = 3.70 \times 10^{-4}$; (5) $p_{adjust} = 5.96 \times 10^{-3}$; (6) $p_{adjust} = 1.44 \times 10^{-4}$; (7) $p_{adjust} = 4.97 \times 10^{-7}$; (8) $p_{adjust} = 1.75 \times 10^{-2}$. **c, d** Data are presented as combined violin and box plots (thick line in the box corresponds to median; lower and upper edges of the box indicating the first (Q1) and third (Q3) quartiles; whiskers extend to 1.5 times the IQR). Individual data points are presented using the jitter method. Kruskal-Wallis tests with Dunn posthoc tests (two-sided) and Benjamini-Hochberg corrections for multiple comparisons were used to test for statistical significance. *, $p_{adjust} < 0.05$; **, $p_{adjust} < 0.01$; and ***, $p_{adjust} < 0.001$. Source data are provided as a Source Data file.

Overall, the CIs observed in the CRCTropism cohort resulted in a characteristic pattern of gains and losses, enriched in whole chromosomal arm aneuploidies and certain focal CNVs (Fig. 1a and Supplementary Fig. 3). In line with previous results[67], the CRCTropism cohort showed the typical CI pattern associated with CRC, including the loss

of the short arm of chromosome 8 accompanied by a gain of the long arm of this chromosome, with typical breakpoints located in chromosomal bands 8p11 or 8p12 (Supplementary Fig. 4).

Focusing on primary CRCs, the most common DNA copy number aberrations were +13q, -18q, and +20q (Fig. 1a). A similar pattern of CIs

was seen in liver, lung, and brain metastases (Fig. 1a). However, notable differences emerged in the frequencies of specific alterations at secondary sites. For example, brain metastases exhibited a higher proportion of gains at 7p (liver 0.433 [log2 ratio -0.48], lung 0.265 [log2 ratio -1.19] vs. brain 0.606; $p_{adjust} = 0.025$; Fisher exact test with Benjamini-Hochberg correction), and 12p (liver 0.067 [log2 ratio -1.85], lung 0.015 [log2 ratio -4.01] vs. brain 0.242; $p_{adjust} = 7.80 \times 10^{-3}$). Moreover, lung metastases showed a significantly lower frequency of +20p (liver 0.267 [log2 ratio -0.18], lung 0.044 [log2 ratio -2.78] vs. brain 0.303; $p_{adjust} = 4.22 \times 10^{-3}$), among other aberrations.

## Chromosomal arm aneuploidies as major source of DNA copy number aberrations

To address the CI profile in detail, a series of analyses was conducted. First, we asked about the type of CI (i.e., chromosomal arm aneuploidies vs. focal subchromosomal arm CNVs). Overall, apparent chromosomal arm aneuploidies contributed as the major source of CIs, accounting for about three-quarters of all CIs (Fig. 1b). Brain metastases exhibited the highest number of apparent chromosomal arm aneuploidies, while lung metastases showed the lowest (Fig. 1c). Specifically, the median number of aneuploidies was 6.5 (interquartile range [IQR] 6) in liver metastases, 3 (IQR 6) in lung metastases, and 8 (IQR 6) in brain metastases. Statistical analysis using the Kruskal-Wallis test revealed significant differences ($p_{adjust} = 1.35 \times 10^{-18}$ with Benjamini-Hochberg correction; Supplementary Table 2). Subsequent pairwise comparisons with Dunn test and Benjamini-Hochberg correction indicated adjusted $p$-values of $2.96 \times 10^{-3}$ for liver vs. lung metastasis, $1.31 \times 10^{-2}$ for liver vs. brain metastasis, and $6.94 \times 10^{-6}$ for lung vs. brain metastasis.

## Profile of whole chromosomal arm and subchromosomal arm copy number aberrations

We next explored the overall pattern of CIs by integrating data from both whole chromosomal arms and subchromosomal regions. Lung metastases had the fewest CIs, with a median count of 5 (IQR 7.25; Fig. 1d), while brain metastases exhibited the highest count, with a median of 13 CIs (IQR 10). Liver metastases fell in between, showing a median of 9 CIs (IQR 9). The Kruskal-Wallis test demonstrated significant differences among these groups ($p_{adjust} = 6.10 \times 10^{-23}$; Supplementary Table 2). Subsequent pairwise comparisons using Dunn test with Benjamini-Hochberg correction indicated that brain metastases had a significantly greater number of CIs compared to both liver metastases ($p_{adjust} = 5.57 \times 10^{-3}$) and lung metastases ($p_{adjust} = 1.37 \times 10^{-6}$). These significant differences in total CI counts were consistent with the observed chromosomal arm aneuploidies.

This pattern was also evident when analyzing both losses and gains separately. The Kruskal-Wallis tests revealed significant findings for both categories ($p_{adjust} = 5.86 \times 10^{-8}$ for losses and $p_{adjust} = 1.86 \times 10^{-5}$ for gains). In particular, liver metastases had a median of 4 losses (IQR 5), lung metastases had a median of 3 losses (IQR 6), while brain metastases had the highest median of 6 losses (IQR 7). For gains, liver metastases showed a median of 4 gains (IQR 5), lung metastases had 1.5 gains in median (IQR 3.25), whereas brain metastases demonstrated a higher median of 7 gains (IQR 4). With the exception of losses in liver vs. lung metastases, all pairwise comparisons yielded statistically significant results in Dunn tests with Benjamini-Hochberg correction (Supplementary Table 2). Thus, we conclude that CRC metastases exhibit distinct variations in the counts of individual CI types.

## Organotropic mapping identifies site-specific CI patterns in CRC metastases

Our analysis of DNA copy number alterations in the cohort suggested distinct patterns of CIs in metastatic CRC dependent on the site of metastasis. To test whether certain CIs are enriched at specific metastatic sites, we used a ternary plot, which highlights significant differences in the proportion of lesions harboring particular imbalances (Fig. 2a). We refer to this visualization as organotropic map. This map projects the frequency of observed CIs into a ternary diagram, with statistical significance assessed by Fisher exact test with Benjamini-Hochberg correction for multiple comparisons. In this representation, the size of each circle corresponds to the frequency of the respective CI within the cohort. The corners of the organotropic map denote the secondary sites. Organotropic CIs are positioned in proximity to one of the corners (i.e., metastasis organs), if they are predominantly associated with a single metastatic site; those enriched at two sites are placed in the middle between two corners. Conversely, genetic changes uniformly associated with all three sites are situated at the center of the organotropic map.

A total of sixteen statistically significant CNVs were identified (Fig. 2a and Supplementary Fig. 5). Notably, none of these organotropic CIs were associated with lung metastasis. Nine CIs were enriched in both liver and brain metastasis, specifically +7p, -8p, +8q, +13q, +16p, +16q, -18p, +20p, and +20q (Fig. 2a,b and Supplementary Fig. 5). These CIs largely correspond to some of the most frequently observed chromosomal aberrations in CRC. In addition, seven organotropic CIs were specifically enriched in brain metastasis (Fig. 2a), with four CIs identified in more than 15% of brain metastases (Supplementary Fig. 5). These CIs include -3q, +5q, -6q, and +12p. Overall, we conclude that CRC exhibits a range of organotropic CIs that are preferentially observed in specific metastatic organs. This organotropic signature of CRC metastases is visually summarized in Fig. 2c, highlighting the chromosomal imbalances across autosomes.

## Gene-Level Mapping of CNVs in CRC Metastases

To address CNVs in an independent cohort, we took advantage of the MSK MetTropism cohort[28], which comprises a total of 3,548 CRC samples including 2,401 primary tumors, 624 liver metastases, 146 lung metastases, and 22 brain/CNS metastases (hereinafter referred to as brain metastases). The demographics of this cohort are given in Supplementary Table 3, and an overview of the analyses performed is provided in Supplementary Fig. 1. The overall pattern of chromosomal arm level gains and losses in the MSK MetTropism cohort was consistent with that observed in the CRCTropism cohort, with +13q, -18, and +20q being the most common aneuploidies (Fig. 3a). However, as observed for the CRCTropism cohort, brain metastases showed a higher number of CIs. Amongst others, -3p, -3q, +5q, +7p, +8q, +12p, and +13q were enriched in brain metastases. In line with these observations, the fraction genome altered (FGA), a measure of the total proportion of the genome affected by CNVs, was significantly higher in brain metastases compared to liver and lung metastases (Kruskal-Wallis test, $p = 8.23 \times 10^{-9}$, followed by Dunn test with Benjamini-Hochberg correction: brain vs. liver, $p_{adjust} = 4.66 \times 10^{-9}$; brain vs. lung, $p_{adjust} = 5.50 \times 10^{-9}$). In contrast, there were no statically significant differences in TMB between the three metastatic sites ($p = 0.113$, Kruskal-Wallis test; Fig. 3b).

To further characterize these genomic alterations, we performed a gene-level analysis of the MSK MetTropism cohort. This analysis revealed a distinct pattern of organotropic CNVs, predominantly enriched in brain metastases – consistent with the higher FGA observed in this group and corroborating the findings from our CRCTropism cohort (Fig. 3c). These enriched genes spanned various functional categories, including the following (note that genes often have multiple cellular functions; for simplicity, each gene has been assigned to only one category):

(I)   signaling: e.g., *PDGFRB* (chromosomal band 5q32), *FGFR4* (5q35.2), *FLT4* (5q35.3), *BRAF* (7q34), *RET* (10q11.21), *KRAS* (12p12.1), *ERBB3* (12q13.2), *FLT3* (13q12.2), *FLT1* (13q12.3), and *MAP2K1* (15q22.31);

(II)  chromatin organization/transcription: e.g., *ARID1A* (1p36.11), *SETD2* (3p21.31), *PBRM1* (3p21.1), *TERT* (5p15.33), *CDK8* (13q12.13), and *CTCF* (16q22.1);

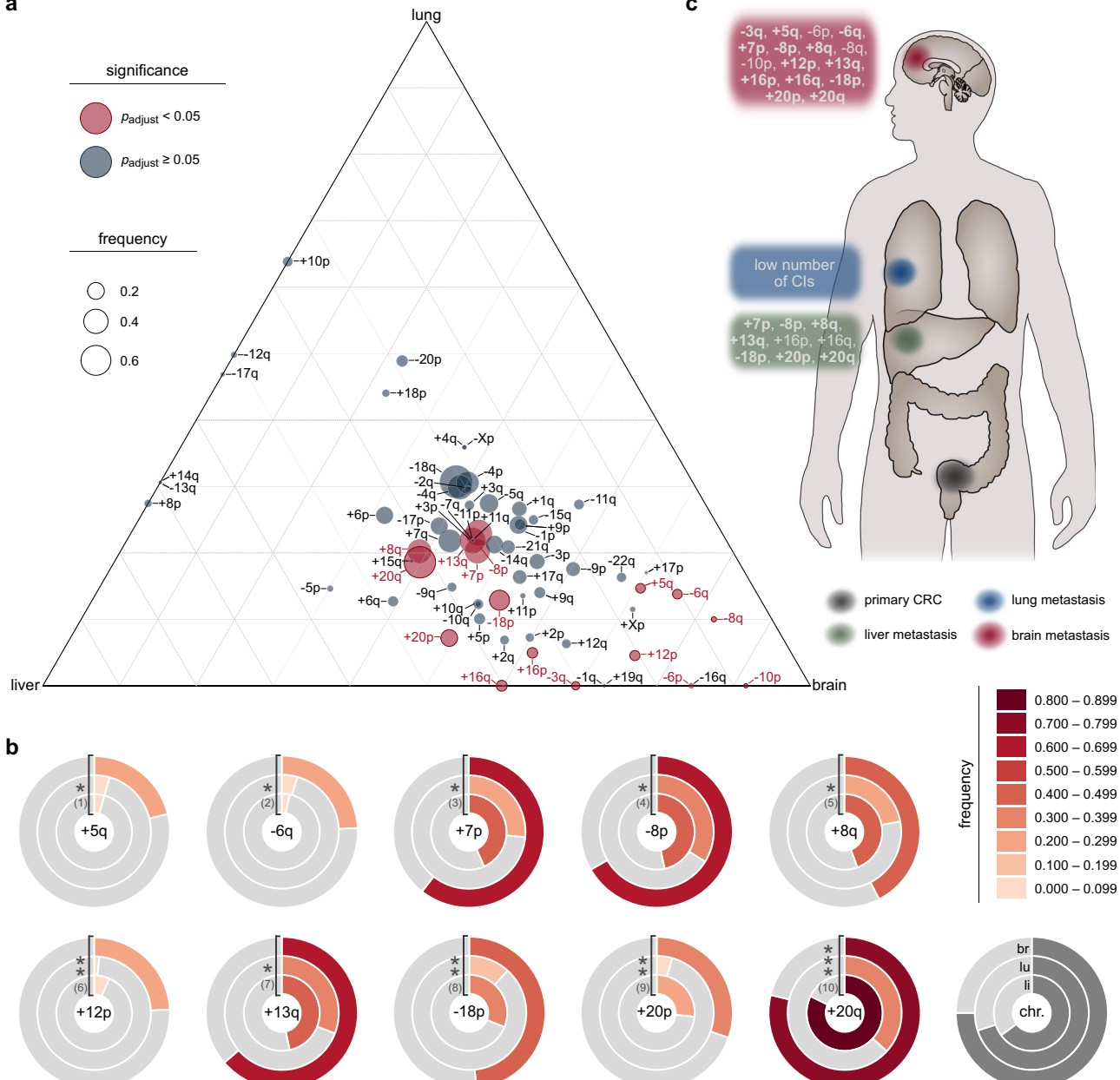

**Fig. 2 | Site-specific pattern of DNA copy number aberrations in CRC metastases based on the CRCTropism cohort. a** Organotropic map presenting CIs with site-specific enrichment ($n = 191$). All $p$-values were obtained by two-sided Fisher exact tests followed by Benjamini-Hochberg adjustment for multiple comparison (legend). The size of the bubbles encodes the frequency a given CI was observed in the cohort (legend). **b** Doughnut plots showing the fraction of metastases ($n = 191$) that harbor selected organotropic CIs (see legend for the assignment of secondary organs: li, liver; lu, lung; br, brain; chr., chromosomal arm; see also Supplementary Fig. 5 for a full representation of all organotropic CIs). The frequency of occurrence is color-encoded according to the legend shown on the right. Two-sided Fisher exact tests with Benjamini-Hochberg adjustment for multiple comparison were used (*, $p_{adjust} < 0.05$; **, $p_{adjust} < 0.01$, and ***, $p_{adjust} < 0.001$). (1) $p_{adjust} = 3.80 \times 10^{-2}$; (2) $p_{adjust} = 1.14 \times 10^{-2}$; (3) $p_{adjust} = 2.48 \times 10^{-2}$; (4) $p_{adjust} = 3.80 \times 10^{-2}$; (5) $p_{adjust} = 4.10 \times 10^{-2}$; (6) $p_{adjust} = 7.80 \times 10^{-3}$; (7) $p_{adjust} = 3.80 \times 10^{-2}$; (8) $p_{adjust} = 4.85 \times 10^{-3}$; (9) $p_{adjust} = 4.22 \times 10^{-3}$; (10) $p_{adjust} = 3.08 \times 10^{-7}$. **c** Graphical summary of autosomal organotropic CIs with arbitrary intra-organ sites of lesions. CIs observed in more than 15% of the respective metastases are highlighted in bold. Source data are provided as a Source Data file.

(III) DNA damage response and repair: e.g., *MSH6* (2p16.3), *BARD1* (2q35), *MLH1* (3p22.2), *BAP1* (3p21.1), *ATR* (3q23), *MSH3* (5q14.1), *MRE11A* (11q21), and *ATM* (11q22.3), *RAD51B* (14q24.1), *BRCA1* (17q21.31), *RAD51C* (17q22), *CHEK2* (22q12.1);

(IV) cell cycle: e.g., *CDK4* (12q14.1), *MDM2* (12q15); and

(V) microRNA biology: e.g., *DROSHA* (5p13.3).

Some organotropic genes, such as *AGO2* (8q24.3), were enriched in both liver and brain metastases. Gains in a single gene, the transcription factor *JUN* (1p32.1), were uniquely enriched in both lung and brain metastases.

## Co-occurrence pattern of organotropic *KRAS* amplifications and mutations

One of the organotropic genes identified in our analysis is *KRAS*, located on 12p12.1, with amplifications significantly enriched in brain metastases. Organotropic mapping of our CRCTropism cohort demonstrated a significant enrichment of +12p in brain metastases

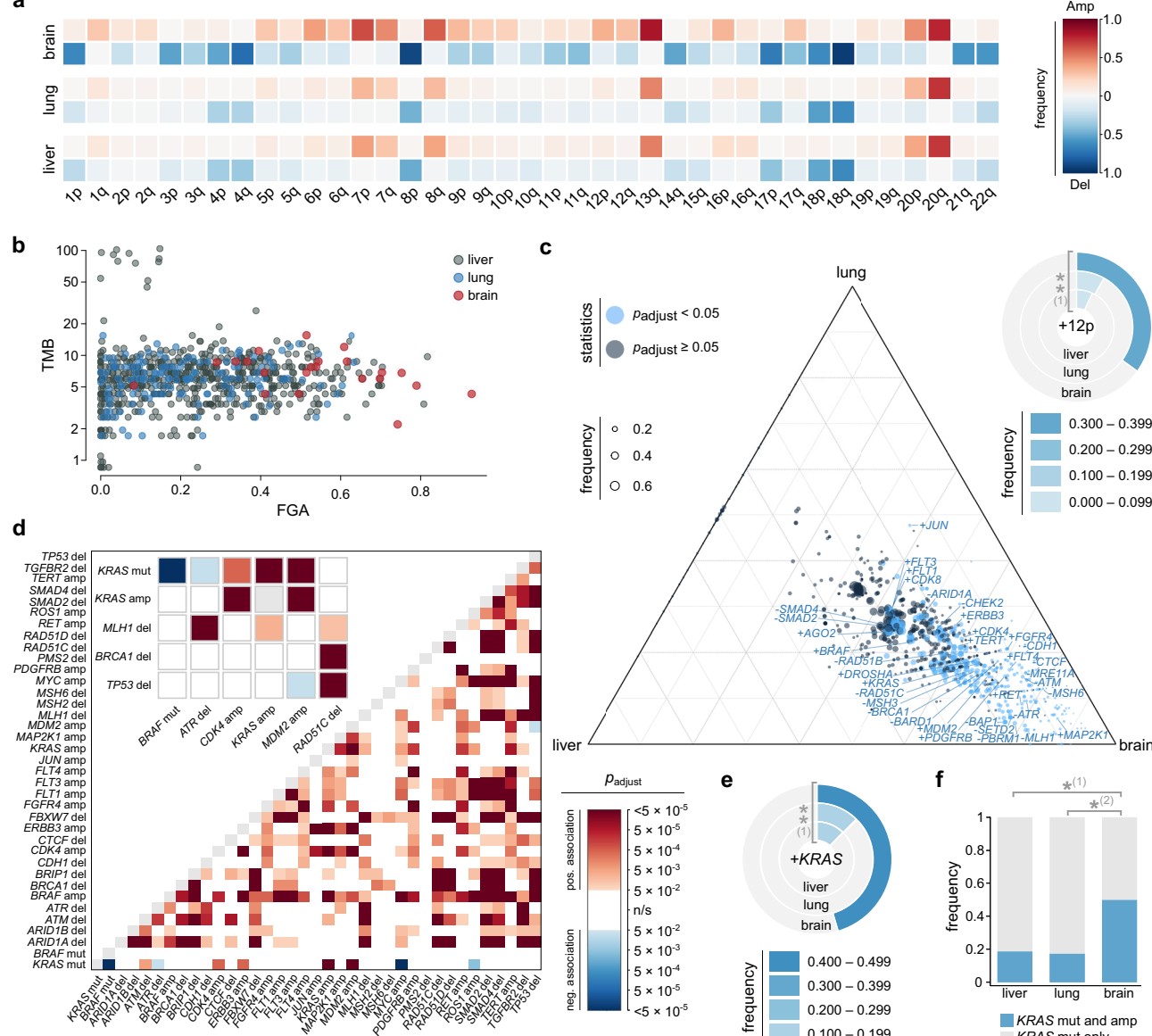

**Fig. 3 | Organotropic genetic signatures of metastatic CRC in the MetTropism cohort. a** Pattern of chromosomal arm aneuploidies in liver, lung, and brain metastases determined using ASCETS, with $n = 792$ metastases subjected to analysis. For each metastatic site, the frequency of the chromosomal arm aneuploidy (bottom, deletion; top, amplification) is color-coded as shown in the color scale at the right. **b** Profile of tumor mutational burden (TMB) and fraction genome altered (FGA) for each liver, lung, and brain metastasis (FGA: $p = 8.23 \times 10^{-9}$ [Kruskal-Wallis test], $p_{adjust} = 4.66 \times 10^{-9}$ for brain metastases vs. liver and $p_{adjust} = 5.50 \times 10^{-9}$ for brain metastases vs. lung metastases [Dunn posthoc test with Benjamini-Hochberg correction]; TMB: $p = 0.113$ [Kruskal-Wallis test]; $n = 792$). **c** Organotropic map of gene-level gains and losses derived from GISTIC 2.0 ($n = 792$). Significant CIs are indicated in blue, with the event frequency represented by the bubble size (legend). The doughnut plot in the top right shows the fraction of metastases with +12p as determined using ASCETS ($n = 758$), where the occurrence frequency is color-

encoded according to the legend on the right. (1) $p_{adjust} = 5.19 \times 10^{-3}$. **d** Co-occurrence probability mapping of genetic changes observed in CRC metastases ($n = 792$), with red shades indicating significant positive associations and blue shades depicting significant negative associations, based on co-occurrence modeling (two-sided), followed by Benjamini-Hochberg adjustment. The inset presents close-ups of selected genetic changes (amp, amplification; del, deletion; mut, mutation). **e** Fraction of metastases ($n = 792$) harboring *KRAS* amplifications determined using GISTIC 2.0 (color-coded according to the legend). (1) $p_{adjust} = 3.49 \times 10^{-3}$. **f** Bar plot depicting the frequency of *KRAS* amplifications among metastases ($n = 352$) with oncogenic *KRAS* mutations. (1) $p_{adjust} = 0.019$; (2) $p_{adjust} = 0.019$. **c**, **e**, **f** Adjusted p-values were obtained by two-sided Fisher exact test with Benjamini-Hochberg adjustment for multiple comparisons. *, $p_{adjust} < 0.05$; **, $p_{adjust} < 0.01$. Source data are provided as a Source Data file.

(Fig. 2a). This finding was independently corroborated in the MSK MetTropism cohort, which also showed a significant enrichment of +12p chromosomal arm aneuploidy in brain metastases (7%, 8%, and 35% for liver, lung and brain metastases, respectively; $p_{adjust} = 5.19 \times 10^{-3}$, Fisher exact test followed by Benjamini-Hochberg correction; Fig. 3c, right).

Given that oncogenic *KRAS* mutations are known drivers in CRC, we next asked whether *KRAS* amplifications are associated with

oncogenic *KRAS* mutations. To this end, we analyzed the co-occurrence of event pairs in the lesions and took advantage of a probabilistic model that accounts for the frequencies of the respective events. Based on this model, positive and negative co-occurrences, defined as two events observed together with a significantly higher or lower frequency, respectively, than assuming a random association, were visualized in a color-encoded triangular heatmap of Benjamini-Hochberg adjusted p-values. A significant positive association between

*KRAS* mutations and *KRAS* amplifications was observed in the co-occurrence analysis ($p_{adjust} < 5 \times 10^{-5}$; Fig. 3d, inset). Notably, *KRAS* gains were identified in nearly half of the brain metastases, but only in about 12% in liver and lung metastases ($p_{adjust} = 3.49 \times 10^{-3}$, Fisher exact test followed by Benjamini-Hochberg correction; Fig. 3e), in line with the analysis of +12p chromosomal arm aneuploidy. Further analysis revealed that the proportion of CRC metastases with simultaneous *KRAS* mutation and amplification was significantly higher in brain metastases compared to liver and lung metastases (liver, 19%; lung, 17%; brain, 50%; $p_{adjust} = 0.019$, each; Fisher exact test with Benjamini-Hochberg correction; Fig. 3f).

Oncogenic *KRAS* mutations were found to co-occur with several other CIs, such as *CDK4* amplifications and *MDM2* amplifications (Fig. 3d, inset). As anticipated, a significant negative association was observed between *KRAS* mutations and *BRAF* mutations, and *ATR* deletions (Fig. 3d, inset). Similarly, a negative association was noted between *TP53* deletions and *MDM2* amplifications. These findings collectively indicate a significant co-occurrence pattern, suggesting a unique genomic landscape that may contribute to CRC metastasis.

## *KRAS* CNVs translate into increased KRAS pathway activity

We hypothesized that the co-occurrence of *KRAS* mutations and amplifications may facilitate metastatic spread through the rewiring of the cancer cell pathways, reflected by changes in gene expression patterns. To test this hypothesis, we analyzed the TCGA cohort, which provides genomic, transcriptomic and proteomic data[65,66,68]. Patient characteristics are given in Supplementary Table 4. We identified 64 CRC cases exhibiting simultaneous *KRAS* mutation and amplification (Fig. 4a). Transcriptome analysis revealed significantly higher *KRAS* expression levels in CRCs with both *KRAS* mutation and amplification compared to those with *KRAS* mutation alone ($p_{adjust} = 1.32 \times 10^{-3}$, Wilcoxon rank sum test with Benjamini-Hochberg correction; Fig. 4b), confirming that the *KRAS* amplification is linked to increased *KRAS* expression.

To test whether the significant increase in *KRAS* expression in *KRAS*-mutated and amplified CRCs is also reflected in pathway activity, we analyzed the levels of activated (phosphorylated) downstream effector proteins mitogen-activated protein kinase kinase 1 and 2 (MAP2K1/2, also known as MEK1 and MEK2) in CRCs harboring oncogenic *KRAS* mutations. This analysis utilized reverse-phase protein array (RPPA) data on 160 CRCs, which provide information on both total and phosphorylated protein levels for selected cancer-associated proteins[68]. Oncogenic KRAS signals through RAF, resulting in the phosphorylation of MAP2K1/2 at two serine residues within the activation loop[69]. We hypothesized that, in addition to KRAS, other factors, including the protein levels of downstream target proteins, might influence the amount of phosphorylated MAP2K1/2. Consequently, we employed multiple linear regression to model MAP2K1/2 phosphorylation.

*KRAS*_CNV level correlated with *KRAS* mRNA expression, necessitating the exclusion of the latter from the model. The remaining variables showed no relevant multicollinearity, with variance inflation factors (VIF) of 1.008 for KRAS_CNV, 1.009 for pBRAF_S445 (a measure of constitutively phosphorylated BRAF), and 1.016 for pMAP2K1 (total protein). Data on total MAP2K2 were not available. Our multiple linear regression analysis confirmed that the *KRAS*_CNV is a significant predictor of phosphorylated MAP2K1/2 ($p = 0.041$, likelihood ratio test). In CRC with oncogenic *KRAS* mutations, *KRAS*_CNV positively contributed to predicting levels of phosphorylated MAP2K1/2, alongside pBRAF_S445 and pMAP2K1, all of which demonstrated similar importance scores (Fig. 4c). These findings suggest that the copy number of *KRAS* translates into increased KRAS pathway activity at the functional protein level.

## Rewiring of CRC cells with simultaneous *KRAS* mutation and amplification

Building on our findings that concurrent *KRAS* mutations and amplifications result in increased pathway activity, we sought to further

investigate the functional consequences of these genetic alterations. Specifically, we hypothesized that the concurrent *KRAS* mutation and amplification may lead to a more accentuated reprogramming of cellular pathways, ultimately influencing the aggressive behavior of CRC. To explore these functional implications, we performed gene set enrichment analysis (GSEA) utilizing oncogenic signature definitions. We were particularly interested in the M2900 gene set, which represents genes upregulated in cancer cells overexpressing an oncogenic *KRAS* form, as it closely reflects our cohort. Comparing CRC samples with both *KRAS* mutation and amplification to those with *KRAS* mutation alone, we observed significant enrichment of this gene set in the former (normalized enrichment score, $NES_{GSEA} = 1.95$, $p < 1 \times 10^{-4}$, $p_{adjust} = 0.040$; Fig. 4d). This finding suggests that *KRAS* mutation and amplification together contribute to a more pronounced *KRAS*-driven oncogenic profile, potentially enhancing the aggressive phenotype of these tumors.

Further analysis of the transcriptome identified 200 significantly upregulated genes and 156 significantly downregulated genes in CRCs with concurrent *KRAS* mutation and amplification. The volcano plot of expressed genes showed a unique peak-like concentration at a fold change of about 1.25 in the CRC group with both *KRAS* mutation and amplification (Fig. 4e). Notably, most upregulated genes shared a chromosomal location on chromosome 12 (Fig. 4e, inset), suggesting that other loci on this chromosome may be co-amplified alongside *KRAS* due to larger CNVs.

To validate our findings, we performed GSEA using C1 gene set definitions from MSigDB, which correspond to chromosomal bands. This analysis confirmed that genes enriched in CRCs with *KRAS* mutation and amplification are predominantly located on both arms of chromosome 12 (Fig. 4f; $NES_{GSEA}$ ranging from 2.80 to 1.95; $p_{adjust} < 9 \times 10^{-5}$ to 0.038), with no other chromosomal bands showing significance. Enriched genes included key oncogenes such as *CDK4* and *MDM2* (both encoded on 12q), further supporting the notion that the peak-like asymmetry in the volcano plot is due to co-amplification of multiple loci on chromosome 12 along with *KRAS*.

To further characterize the functional landscape of the differentially expressed genes, we first tested the upregulated genes, independent of their chromosomal location, for enrichment using Enrichr with two complementary gene set definitions, Gene Ontology (GO) and MSigDB Hallmark. This analysis identified several significant biological processes, including 'Positive Regulation Of T-helper 17 Cell Differentiation' (GO:2000321; $p_{adjust} = 0.019$), 'Negative Regulation Of Peptidase Activity' (GO:0010466; $p_{adjust} = 0.027$), 'Regulation Of Cell Population Proliferation' (GO:0042127; $p_{adjust} = 0.042$), and 'Carbohydrate Catabolic Process' (GO:0016052; $p_{adjust} = 0.042$; Fig. 4g).

We further analyzed the upregulated genes separately according to their chromosomal location, performing separate Enrichr analyses for genes on chromosome 12 and those outside of it using MSigDB Hallmark definitions. The enriched gene set for differentially expressed genes on chromosome 12 was the 'p53 Pathway' that includes *MDM2* ($p_{adjust} = 8.18 \times 10^{-3}$), while those outside chromosome 12 included 'Angiogenesis' ($p_{adjust} = 2.87 \times 10^{-3}$), 'Inflammatory Response' ($p_{adjust} = 2.87 \times 10^{-3}$), 'Epithelial Mesenchymal Transition' (EMT; $p_{adjust} = 2.87 \times 10^{-3}$) and 'Apical Junction' ($p_{adjust} = 0.014$; all *p*-values adjusted the Benjamini-Hochberg method for multiple testing). These biological functions are enriched in processes associated with hypoxia[70–72].

To investigate this association, we examined whether CRCs with *KRAS* mutation and amplification demonstrate enhanced hypoxia adaptation by analyzing the Winter hypoxia score as a measure of hypoxic response[73]. Our results indicate that CRCs with concurrent *KRAS* mutation and amplification exhibited significantly higher hypoxia scores compared to those with *KRAS* mutation alone ($p = 8.75 \times 10^{-4}$, Wilcoxon rank sum test; Fig. 4h).

Given our observation of increased hypoxia adaptation in *KRAS* mutated and amplified CRC, we explored the potential implications of

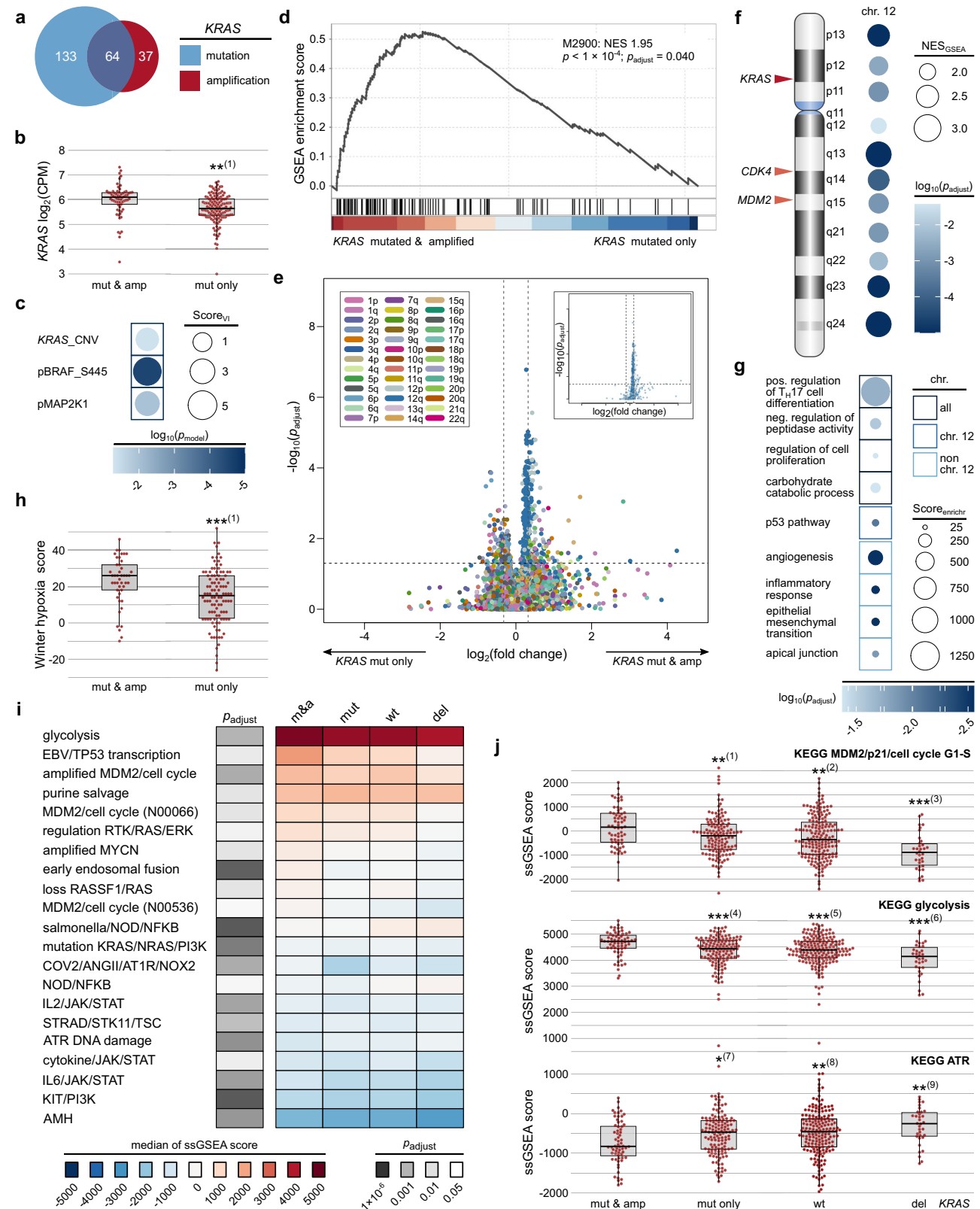

these alterations on hypoxia-related pathways, complemented by additional medically relevant pathways as implemented in the Kyoto Encyclopedia of Genes and Genomes (KEGG) MEDICUS gene set definitions. We hypothesized that highly *KRAS*-responsive gene sets would show a systematic increase or decrease in enrichments across the following groups: (1) CRC with simultaneous *KRAS* mutation and amplification, (2) CRC with *KRAS* mutation only, (3) CRC with wild-type

*KRAS*, and (4) CRC with *KRAS* deletion (no *KRAS* mutation). To test this hypothesis, we performed single-sample gene set enrichment analysis (ssGSEA), which calculates an enrichment score for each sample-gene set pairing. In total, 94 gene sets were statistically significant using the Kruskal-Wallis test with Benjamini-Hochberg correction. Of these, 21 showed statistically significant differences in Dunn test with Benjamini-Hochberg correction when comparing CRC with *KRAS*

**Fig. 4 | Rewiring of *KRAS* mutated and amplified CRCs as indicated by the TCGA cohort. a** Venn diagram showing the relationship between CRCs with *KRAS* mutations (blue circle) and amplifications (red circles). **b** Box plot of normalized *KRAS* expression levels, with *p*-values from two-sided Wilcoxon rank-sum test and Benjamini-Hochberg adjustment for multiple comparison ($n = 64$ and $n = 133$ for CRC with *KRAS* mutation and amplification and CRC with *KRAS* mutation, respectively). (1) $p_{\text{adjust}} = 1.32 \times 10^{-3}$. **c** Multiple linear regression indicating *KRAS*_CNV as a significant predictor of phosphorylated MAP2K1/2, along with pBRAF_S445, and pMAP2K1 with similar variable importance scores (Score$_{VI}$), with $n = 160$. **d.** Two-sided GSEA using the M2900 gene set, derived from cells overexpressing oncogenic *KRAS*. The normalized enrichment score (NES), nominal *p*-value and false discovery rate (FDR)-adjusted *p*-value are shown ($n = 197$). **e** Volcano plot of differential gene expression, colored by chromosomal arm (see legend). The inset highlights genes on chromosome 12. Statistical significance was assessed using two-sided Wilcoxon rank-sum tests with Benjamini-Hochberg correction for multiple comparisons ($n = 197$). The horizontal dashed line indicates $p_{\text{adjust}} = 0.05$, while vertical dashed lines correspond to a 1.25-fold change. **f** Two-sided GSEA using C1 positional gene sets on chromosomal bands enriched in *KRAS* mutated and amplified CRCs vs. CRC with *KRAS* mutation ($n = 197$); circle size indicates NES and color the FDR-adjusted *p*-value. An ideogram of chromosome 12 is provided for orientation. **g** Enrichr analysis of upregulated genes in CRCs with *KRAS* mutation and amplification vs. CRC with *KRAS* mutation ($n = 197$),

using enriched genes on chromosome 12 and other chromosomes. Circle size reflects the Enrichr score and color indicates the Benjamini-Hochberg-adjusted *p*-value. The Gene Ontology and MSigDB Hallmark genes sets were tested. **h.** Winter hypoxia score comparison using two-sided Wilcoxon rank-sum test ($n = 45$ and $n = 106$ for CRC with *KRAS* mutation and amplification and CRC with *KRAS* mutation, respectively; (1) $p_{\text{adjust}} = 8.75 \times 10^{-4}$. **i** Heatmap of median ssGSEA scores using KEGG_MEDICUS gene sets, with Benjamini-Hochberg adjusted *p*-values from Kruskal-Wallis tests encoded in grayscale ($n = 412$). m&a, *KRAS* mutated and amplified CRCs; mut, *KRAS* mutated CRCs; wt, *KRAS* wildtype CRCs; del, *KRAS* deletion CRCs (without *KRAS* mutation). **j** Box plots of ssGSEA scores for selected gene sets, analyzed by Kruskal-Wallis tests with Dunn posthoc test (two-sided) and Benjamini-Hochberg adjustment ($n = 64$ for *KRAS* mutated and amplified CRCs; $n = 133$ for *KRAS* mutated CRCs; $n = 181$ for *KRAS* wildtype CRCs; $n = 34$ for *KRAS* deletion CRCs (without *KRAS* mutation); (1) $p_{\text{adjust}} = 6.78 \times 10^{-3}$; (2) $p_{\text{adjust}} = 1.69 \times 10^{-3}$; (3) $p_{\text{adjust}} = 1.59 \times 10^{-7}$; (4) $p_{\text{adjust}} = 6.19 \times 10^{-4}$; (5) $p_{\text{adjust}} = 5.27 \times 10^{-4}$; (6) $p_{\text{adjust}} = 3.20 \times 10^{-6}$; (7) $p_{\text{adjust}} = 3.12 \times 10^{-2}$; (8) $p_{\text{adjust}} = 7.63 \times 10^{-3}$; (9) $p_{\text{adjust}} = 1.17 \times 10^{-3}$). Analyses compare CRCs with *KRAS* mutation and amplification to those with *KRAS* mutation alone (**b, d-h**) as well as to wildtype CRC and CRC with *KRAS* deletions without *KRAS* mutation (**i, j**). (**b, h, j**) Data are presented with individual data points and a box-and-whisker-plot (thick line, median; box edges, Q1/Q3; whiskers, 1.5 × IQR). *, $p_{\text{adjust}} < 0.05$; **, $p_{\text{adjust}} < 0.01$, and ***, $p_{\text{adjust}} < 0.001$. Source data are provided as a Source Data file.

mutation and amplification to CRC with *KRAS* mutation alone (Fig. 4i). The identified pathways and processes included, among others, critical signaling cascades, cell cycle regulation, transcription, DNA damage response, as well as glycolysis, a key metabolic response to hypoxia.

Notably, some pathways exhibited a significant gradual increase or decrease across the four groups, with *KRAS* mutated and amplified CRCs consistently showing the highest or lowest enrichment (Fig. 4j). These pathways included glycolysis (Kruskal-Wallis test $p_{\text{adjust}} = 2.33 \times 10^{-4}$; Dunn test $p_{\text{adjust}} = 6.19 \times 10^{-4}$ [CRC with *KRAS* mutation and amplification vs. CRC with *KRAS* mutation], $5.27 \times 10^{-4}$ [CRC with *KRAS* mutation and amplification vs. wildtype *KRAS* CRC], and $3.20 \times 10^{-6}$ [CRC with *KRAS* mutation and amplification vs. CRC with *KRAS* deletion]), MDM2/P21/cell cycle G1-S (Kruskal-Wallis test $p_{\text{adjust}} = 5.61 \times 10^{-5}$; Dunn test $p_{\text{adjust}} = 6.78 \times 10^{-3}$, $1.69 \times 10^{-3}$, and $1.59 \times 10^{-7}$, respectively), and ATR signaling (Kruskal-Wallis test $p_{\text{adjust}} = 1.32 \times 10^{-2}$; Dunn test $p_{\text{adjust}} = 3.12 \times 10^{-2}$, $7.63 \times 10^{-3}$, and $1.17 \times 10^{-3}$, respectively), among others. Collectively, these results indicate that CRCs with both *KRAS* mutation and amplification exhibit a distinct gene expression pattern characterized by enhanced glycolytic activity, disrupted cell cycle control, and impaired DNA damage response, among other factors, which may, at least in part, be facilitated by co-amplified loci on chromosome 12.

## Upregulation of *HOX* genes
Having established the distinct molecular landscape associated with CRCs exhibiting concurrent *KRAS* mutation and amplification, we next explored other genetic alterations implicated in metastatic organotropism, specifically focusing on CRCs with deletions of the DNA repair genes *MLH1* and *BRCA1*, respectively, both of which exhibit enrichment in brain metastases (Fig. 3c). For *MLH1* deletion CRCs, we identified 18 significantly upregulated and 209 significantly downregulated genes in comparison to control CRC without microsatellite instability (MSI) phenotype (*MLH1* control CRC). The volcano plot revealed a distinct asymmetric pattern linked to downregulated genes on the 3p arm, where *MLH1* is located (Fig. 5a) – similar to the pattern observed in CRCs with *KRAS* mutation and amplification (Fig. 4e). GSEA confirmed significant enrichment of genes located in bands 3p14 to 3p26 that are downregulated in CRC with *MLH1* deletion (Fig. 5b; NES$_{\text{GSEA}}$ ranging from -2.88 to -2.15; $p_{\text{adjust}} < 10^{-3}$ to $3.42 \times 10^{-3}$).

However, we also identified several genes that are upregulated in CRCs with *MLH1* deletion. A g:Profiler analysis of these differentially upregulated genes showed significant enrichment of pathways related to DNA-binding transcription factor activity, embryonic morphogenesis,

and chromatin dynamics (Fig. 5c). In alignment with our g:Profiler findings, differentially expressed genes from the *HOX* family, including *HOXA1* (located on 7p15.2) and *HOXD9* to *HOXD13* (located on 2q31.1), were significantly upregulated in *MLH1*-deleted CRCs, along with *CREB5* (Fig. 5d). Further analysis of the copy number at the *HOXA* and *HOXD* loci indicated CNVs are not the primary cause of the enhanced expression of these genes ($p_{\text{adjust}} = 0.757$ and $0.941$, respectively; Fisher exact test with Benjamini-Hochberg correction; Fig. 5e). This suggests that the observed upregulation of *HOX* gene expression in CRCs with *MLH1* deletion is likely attributable to transcriptional regulation rather than alterations in gene dosage.

In our analysis of CRCs with *BRCA1* deletion, we identified 152 significantly upregulated and 313 significantly downregulated genes compared to control CRCs. The volcano plot revealed a similar peak-like pattern associated with genes on chromosome 17q, where *BRCA1* is located (Fig. 5f). GSEA confirmed significant downregulation of genes encoded on 17q, including other homologous recombination repair (HRR) genes such as *RAD51C*, *RAD51D*, and *BRIP1* (Fig. 5g; NES$_{\text{GSEA}}$ ranging from −2.63 to −2.04; $p_{\text{adjust}} < 5 \times 10^{-5}$ to 0.019). A g:Profiler analysis of the upregulated genes revealed significant enrichment in pathways involved in transcriptional regulation, gene silencing, chromosome condensation, WNT signaling, and activation of *HOX* genes during differentiation (Fig. 5h). Specifically, we observed significant upregulation of *HOXA3* and *HOXA10*, alongside a notable downregulation of *SNX10* (Fig. 5i). Overall, our findings highlight complex molecular alterations associated with CRCs harboring deletions of *MLH1* and *BRCA1*, respectively, and point towards an activation of *HOX* genes in these cancers.

## Oncogenetic modeling identifies sets of early and late cytogenetic events in CRC
The definition of site-specific CIs (Figs. 2a, 3c) suggests that CRCs may share common patterns of cytogenetic evolution, despite the inherent individuality of each tumor's evolution. We therefore reconstructed the cytogenetic evolution of CRC, utilizing two approaches: (I) reconstruction of oncogenetic trees using maximum likelihood estimation, a cross-sectional approach to model common phylogenetic relationships within the entire cohort or its subsets (Fig. 6) and (II) phylogenetic tree reconstruction for individual patients through maximum parsimony using multi-sampling (Fig. 7 and Supplementary Fig. 6).

Oncogenetic modeling was applied to the three metastatic subgroups studied here. Events were classified into nine categories,

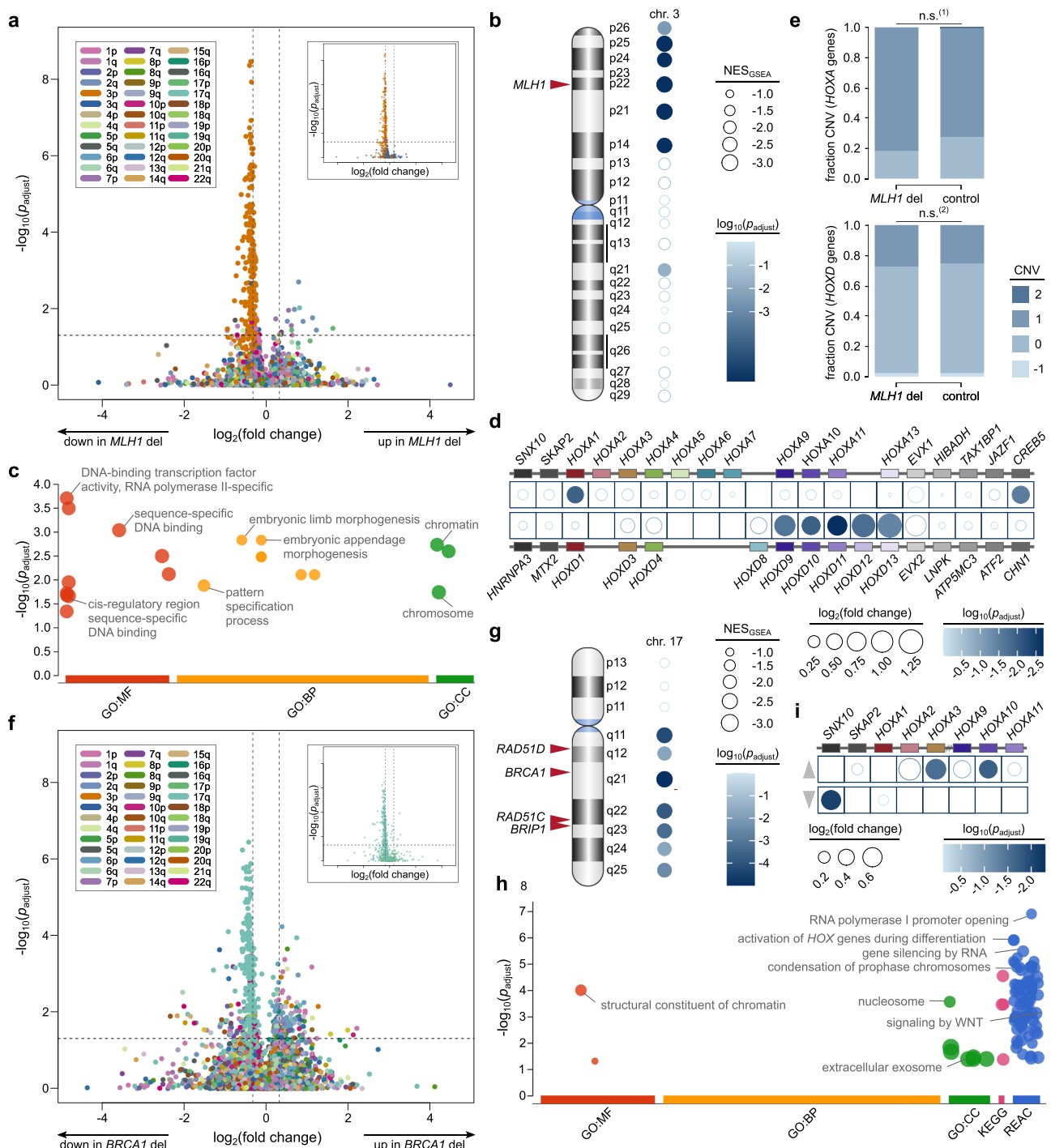

ranging from class 1 (initial events) to class 9 (late events), based on their distance from the root of the oncogenetic tree. All metastatic sites shared the loss of 18q as an initial event (Fig. 6). Likewise, losses of 8p and gains of 13q and 20q were among the earliest events in cytogenetic tumor evolution of CRC, regardless of the metastatic site. In contrast, losses of 1p, 5q, and 14q, amongst others, were positioned further away from the root in the oncogenetic trees, indicating their occurrence as intermediate to late events in cancer development.

The oncogenetic models indicated site-specific differences in the relative timing and nature of cytogenetic events. For example, gain of 20p was modeled as intermediate event in liver and brain metastasis, whereas this event represented a late event in lung metastasis. Conversely, loss of 20p was modeled as an intermediate event in lung metastases but late event in liver metastases. Organotropic CI enriched

in the brain, such as -3q, +5q, -6q, and +12p, were modeled to be acquired later during tumor evolution. In contrast, those enriched in both liver and brain metastases, including +7p, -8p, +8q, +13q, -18p, +20q, were predominantly positioned early in the oncogenetic tree. Thus, the CI patterns observed in the CRC metastasis are also reflected in cytogenetic evolutionary pathways, revealing both common and distinct patterns across different metastatic sites.

## Cytogenetic tumor evolution in individual patients

Given the individual nature of tumor evolution, we explored whether the event divisions shown in the oncogenetic trees—reflecting shared patterns across the cohort—are supported by phylogenetic trees derived from multiple samples of individual patients. To this end, we reconstructed phylogenetic trees for individual patients with multiple

**Fig. 5 | Transcriptional profile of CRCs with deletions of *MLH1* and *BRCA1* in the TCGA cohort. a** Volcano plot showing differential gene expression between *MLH1* deletion CRC and *MLH1* control CRCs, with genes colored by chromosomal arm (inset highlights genes on chromosome 3; horizontal dashed line, $p_{adjust}$ = 0.05; vertical dashed lines, 1.25-fold change). Significance was determined by two-sided Wilcoxon rank-sum tests with Benjamini-Hochberg adjustment (*n* = 276). **b** Two-sided GSEA using C1 positional gene sets indicating that genes encoded on chromosomal arm 3p (ideogram on the left) are significantly downregulated in CRC with *MLH1* deletion compared to *MLH1* control CRCs (*n* = 276). Circle size represents NES, color indicates FDR-adjusted *p*-value. **c** G:profiler analysis of genes upregulated in *MLH1* deletion CRCs, categorized by Gene Ontology (GO), including molecular function (MF), biological process (BP), and cellular component (CC) (*n* = 276). *P*-values are adjusted using the g:SCS method. **d** Differential gene expression of the *HOXA* and *HOXD* loci and flanking genes, with circle size based on log$_2$(fold change) and color reflecting Benjamini-Hochberg-adjusted *p*-values from two-sided Wilcoxon rank-sum tests (open circles indicate $p_{adjust}$ > 0.05; *n* = 276). **e** CNV analysis of the *HOXA* and *HOXD* loci (two-sided Fisher exact test with Benjamini-Hochberg correction; n.s. not significant, (1) $p_{adjust}$ = 0.757; (2)

$p_{adjust}$ = 0.941; *n* = 276)). **f** Volcano plot visualizing differential gene expression for *BRCA1* deletion CRC vs. *BRCA1* control CRC (*n* = 213). Inset shows genes on chromosome 17. Significance determined by two-sided Wilcoxon rank-sum tests with Benjamini-Hochberg adjustment. **g** Two-sided GSEA using C1 positional gene sets for chromosome 17 (ideogram) reveals that genes encoded on 17q are significantly downregulated in CRCs with *BRCA1* deletion compared to *BRCA1* control CRC (*n* = 213). Circle size represents NES, color indicates FDR-adjusted *p*-value. **h** G:profiler analysis of genes upregulated in *BRCA1* deletion CRCs compared to *BRCA1* control CRC (*n* = 213) with GO, KEGG (Kyoto Encyclopedia of Genes and Genomes), and Reactome (REAC) pathways (*p*-values adjusted by g:SCS method). **i** Differential gene expression of the *HOXA* locus and surrounding genes, with the top showing upregulated genes in *BRCA1* deletion CRCs and the bottom showing downregulated ones; circle size reflects log$_2$(fold change), while color indicates Benjamini-Hochberg-adjusted *p*-values from two-sided Wilcoxon rank-sum test (open circles denote $p_{adjust}$ > 0.05; *n* = 213). Panels **a**–**e** compare CRCs with *MLH1* deletion to control CRCs, while (**f**–**i**) compare CRCs with *BRCA1* deletion to control CRCs. Source data are provided as a Source Data file.

tumor samples available for multi-regional cytogenetic analysis (Fig. 7 and Supplementary Fig. 6). The coloring of events was based on the event classes introduced in Fig. 6.

The inferred phylogenetic trees suggested that cytogenetic divergence between the primary CRC and its metastatic spread may occur at various time points (e.g., Figure 7c, g, h). In line with the oncogenetic models (Fig. 6), the loss of the long arm of chromosome 18 is shared among different tumors as a truncal chromosomal aberration in the individual patients' phylogenetic trees (Fig. 7), independent of the site of metastasis. Brain-tropic CIs identified as late events in oncogenetic modeling (Fig. 6) were preferentially seen in the branches of the phylogenetic trees of individual patients, supporting the overall tumor evolution pattern.

To quantify these observations, we divided each phylogenetic tree into trunk/early, branches/intermediate, and sub-branches/late categories and determined a CI score for each individual patient. The analysis revealed that events positioned in the trunk of the individual phylogenetic trees were enriched in events classified as early in the oncogenetic models, while events in the branches were enriched in those attributed as later events in the oncogenetic trees. To assess the statistical significance of these observations, we employed linear mixed-effect modeling, followed by post-hoc testing. This model confirmed significant differences in CI score distributions between the categories ($p = 3.14 \times 10^{-5}$, likelihood ratio test; $p_{adjust} = 9.27 \times 10^{-3}$ for trunk/early vs. branches/intermediate; $p_{adjust} = 9.27 \times 10^{-3}$ branches/intermediate vs. sub-branches/late; and $p_{adjust} = 2.86 \times 10^{-7}$ trunk/early vs. sub-branches/late, posthoc test with Benjamini-Hochberg correction; Supplementary Fig. 7). Thus, while the evolution of each tumor is inherently individual, our analyses suggest that specific chromosomal aberrations are preferentially observed at distinct stages of tumor evolution, including early, intermediate, and late phases.

## Discussion

Our study shows how site-enriched patterns of DNA copy number aberrations shape metastatic behavior, providing insights into CRC progression. Collectively, our findings demonstrate that the DNA copy number aberrations are not randomly distributed; rather, they exhibit distinct patterns at different metastatic sites in terms of (I) the total number of CIs, (II) the frequency and specificity of CNVs, and (III) the sequential order in which specific CIs are acquired during tumor evolution. The organotropic CIs we identified showed significant co-occurrences in probabilistic models. Such co-occurrences would in principle allow for additive or even synergistic effects in promoting metastatic spread.

Most organotropic CIs we identified were either brain-and-liver-tropic or brain-tropic. In this respect, metastatic colonization of the brain poses a number of challenges, as cancer cells must overcome barriers such as the blood-brain barrier and adapt to the unique tissue environment characterized by hypoxia and reduced glucose concentrations[74]. To successfully establish tumors in the brain, metastatic cells likely require additional capabilities beyond those needed for primary tumor growth and peripheral metastasis. The organotropic CIs identified in our analysis correspond to larger CNVs, often involving whole chromosomal arm or whole chromosomal aneuploidies, that impact numerous genes simultaneously. Such extensive alterations may disrupt or deregulate multiple critical regulatory networks, potentially granting cancer cells a selective advantage in adapting to various pressures[75]. In particular, it has been shown that aneuploidy inversely correlates with immune infiltration of tumor tissue[29,75], and chromosomal copy number changes affect immune cell function[76]. These observations suggest that aneuploidy facilitates escape from immune surveillance[29,75]. In addition, aneuploidy appears to modify drug response in cancer cells[77].

One of the organotropic CIs identified in our study is the gain of chromosome 12p, which we modeled as a late event in oncogenetic progression and found to be preferentially associated with brain metastases. This chromosomal arm encodes several critical genes, including the oncogene *KRAS*, which is a driver in CRC development[33]. A recent meta-analysis indicated that the majority of brain metastases examined harbor *KRAS* mutations[48]. Furthermore, previous studies on CRCs have established an association between gains of 12p and somatic *KRAS* mutations[78], reinforcing our observations. This suggests that the +12p event – particularly in the presence of oncogenic *KRAS* mutations – contributes to the metastatic organotropism observed in brain metastases.

At the level of protein function, our analysis shows that the *KRAS* CNV in *KRAS* mutant CRCs translates into enhanced KRAS pathway activity. This finding aligns with our observation showing that *KRAS* amplifications further intensify metabolic rewiring in *KRAS* mutated CRCs, leading to an upregulation of the glycolytic pathway, amongst others. Although glycolysis is less efficient than oxidative phosphorylation in terms of ATP production, it generates crucial biomolecular intermediates necessary for cell proliferation, thereby supporting tumor growth[79,80].

A pivotal player in this metabolic adaptation is Hypoxia Inducible Factor 1 Alpha (HIF1α), a master regulator of glycolysis[81] that upregulates the expression of glycolytic enzymes[82]. In this respect, mutant KRAS enhances HIF1α expression at the transcriptional level[83] and also stabilizes HIF1α by inhibiting prolyl hydroxylases (PHDs), which mark HIF1α for degradation[84]. This dual mechanism not only amplifies glycolytic flux but also promotes a tumor microenvironment conducive to survival under hypoxic conditions.

This adaptative capacity appears to be particularly relevant in the context of brain metastases, as previous studies have shown that HIF1α levels are significantly elevated in CRC brain metastases compared to paired primary CRC samples[85]. In line with increased HIF1α in brain metastases, we observed a significantly higher Winter hypoxia score in *KRAS* mutated and amplified CRC, suggesting that these tumors possess an enhanced capacity for hypoxic adaptation. Given that the partial pressure of oxygen (PtO2) in the human brain is relatively low – estimated at ~20–25 mmHg – compared to around 55–60 mmHg in the large bowel, 50–55 mmHg in the liver, and over 100 mmHg in the lung alveolus[86], we propose that the enhanced hypoxic adaptation in *KRAS* mutated and amplified CRC may facilitate colonization and growth of cancer cells in the brain, thereby contributing to the organotropism towards the brain.

Moreover, oncogene activation such as resulting from *RAS* mutations are well-established inducers of replication stress, which in turn activates the ATR DNA damage response pathway[87]. This stress response serves as an adaptive mechanism, enabling *KRAS*-driven cancer cells to tolerate the genomic instability associated with aberrant proliferation. However, our findings appear to reveal a seemingly paradoxical relationship between mutated and amplified *KRAS* and the ATR damage response pathway in CRC. In particular, our analysis revealed that the ATR pathway is modestly yet significantly downregulated in CRC harboring *KRAS* mutations and amplifications. The extent of this ATR pathway suppression may be a crucial factor influencing the behavior of *KRAS*-mutant cancer cells. When ATR levels are strongly reduced, the resultant genomic instability can lead to synthetic lethality, wherein the accumulation of unrepaired DNA damage triggers cell death in these cancer cells[88]. Conversely, when ATR suppression is moderate, genomic instability may promote tumor development in *KRAS*-mutant cells[88]. In this scenario, residual ATR activity may provide sufficient capacity for DNA repair, allowing the cells to tolerate chromosomal aberrations and eventually benefit from additional driver alterations for continuation of their malignant progression. This delicate balance between DNA damage and repair capacity may drive the evolution of more aggressive cancer phenotypes. Consistently, we observed the highest number of CIs and the greatest FGA in brain metastases.

Organotropism in cancer is likely influenced by a variety of factors, with *KRAS* alterations being a significant contributor. Of particular interest are *MDM2* and *CDK4*, both of which, like *KRAS*, are encoded on chromosome 12. We observed a significant enrichment of gains in *MDM2* and *CDK4* in brain metastases, suggesting that these alterations may play a role in organotropism alongside *KRAS* mutations. This observation is further supported by our pathway analyses, which reveal a notable upregulation of the *MDM2* pathway in CRC with *KRAS* mutations and amplifications. The amplification of *MDM2* leads to dysregulation of the p53 pathway[89], while *CDK4* amplification enhances cell proliferation[90]. Collectively, these genomic alterations may contribute to the aggressive phenotype observed in brain metastases.

In addition to pathways directly linked to genes encompassed by CIs, our analysis uncovered also alterations in other biological processes, including those involving gene members encoded in regions not covered by CIs. Notably, we found that deletions of the DNA repair genes *MLH1* and *BRCA1* are organotropically enriched in brain metastases. While haploinsufficiency of these genes may contribute to genomic instability[91,92], we also observed significant upregulations of specific *HOX* gene sets. In CRCs with *MLH1* deletions, there was significant upregulation of *HOXA1* and *HOXD9 – HOXD13*, whereas in instances of *BRCA1* deletions, we identified increased expression of *HOXA3* and *HOXA10*. This suggests a complex regulatory landscape, in which the transcriptional activation of *HOX* genes may function as a metastasis promoting mechanism, operating independent of apparent gene dosage effects.

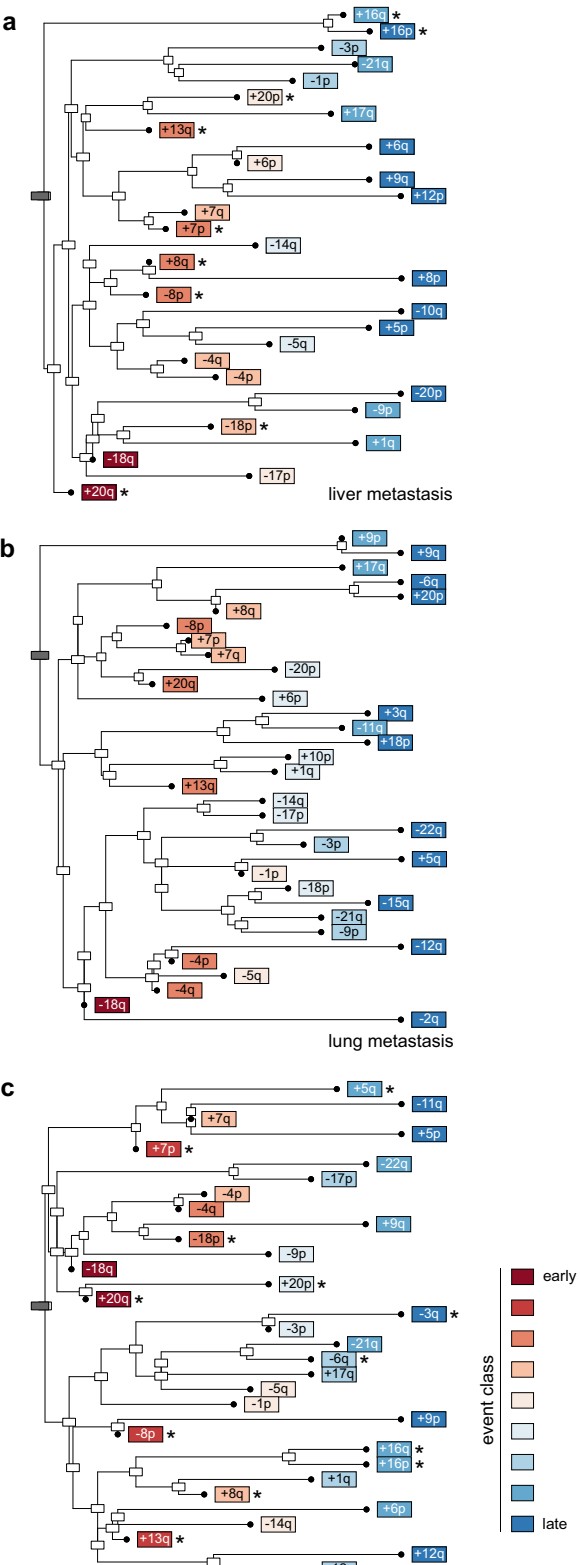

Fig. 6 | Oncogenetic tree models of metastatic CRC based on maximum likelihood estimation. Analyses were done using the CRCTropism cohort for the following metastasis sites: **a**. liver (*n* = 98), **b**. lung (*n* = 69), and **c**. brain (*n* = 33). CIs were classified into nine categories from early to late according to proximity to the root (gray boxes). Organotropic events are marked by asterisks. Source data are provided as a Source Data file.

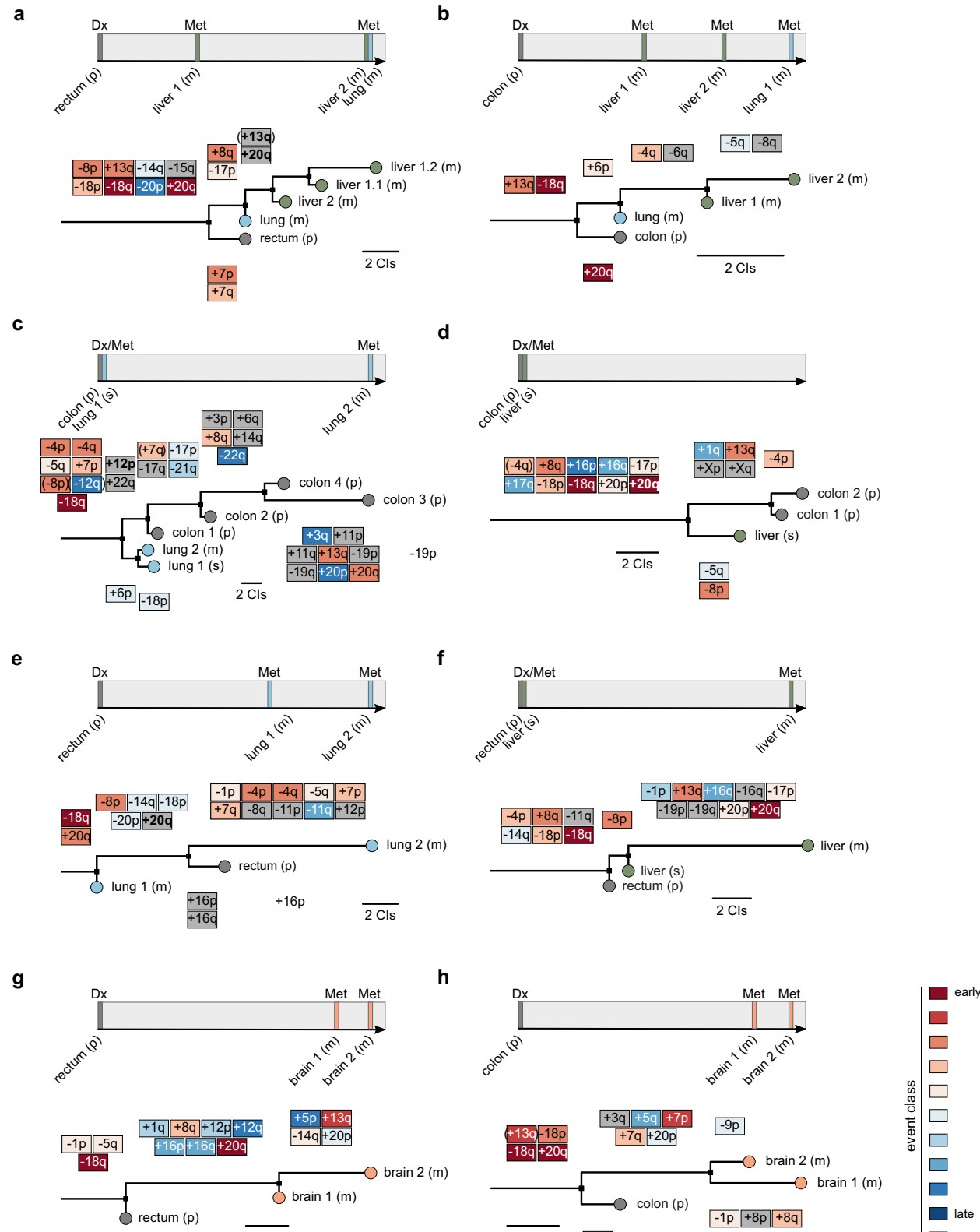

**Fig. 7 | Reconstruction of phylogenetic trees using multi-sample cytogenetic data from individual CRC patients of the CRCTropism cohort. a–h** The clinical course (top) and the reconstructed phylogenetic tree (bottom) are shown, with the tree scaled according to the number of CIs. Maximum parsimony was used for tree inference. CI events are colored based on the classification introduced in Fig. 6. If a particular event could not be uniquely linked (indicated by brackets), it is displayed at the earliest suggested occurrence. High-level amplifications are indicated in bold. The site of the primary tumor (p) and metastases (m, metachronous metastasis; s, synchronous metastasis) is also listed. Dx, diagnosis of primary cancer; Met, metastasis. Source data are provided as a Source Data file.

The *HOX* gene family, known for its role in embryonic development, appears to be repurposed in the context of tumorigenesis. A growing body of literature highlights the involvement of *HOX* genes in orchestrating aggressive tumor phenotypes and enhancing metastatic capabilities across various malignancies. For instance, HOXD9 has been shown to significantly enhance critical malignant characteristics of CRC cells, including proliferation as well as migration and invasion[93]. The authors revealed a bidirectional relationship wherein TGF-β1 elevates *HOXD9* expression, while HOXD9 promotes TGF-β1-triggered EMT, a pivotal process that empowers cancer cells to invade and metastasize[93]. Similarly, HOXA10 has been implicated in promoting EMT through the TGF-β signaling cascade[94], and HOXD13 is linked to proliferation and invasion[95], reinforcing the notion that *HOX* gene dysregulation may contribute to the aggressive behavior of CRC.

In contrast to liver and brain metastases, the only lung-enriched CNV is a gain of *JUN*, which is also enriched in brain metastases. *JUN* codes for a pro-proliferative transcription factor[96] that interacts with the RAS signaling pathway[97]. This seems remarkable given the previously reported increased rate of *KRAS* mutations in CRC lung metastases[98]. The marked underrepresentation of organotropic events in lung metastases is in line with a recent study[45], and it has also been reported that *SRC*, localized in chromosomal band 20q11.23, is less frequently amplified in lung metastases[28]. Overall, lung metastases showed the lowest number of CIs in our study. Taken together, these observations suggest that site-specificity in pulmonary seeding – besides a potential role of the JUN pathway – may follow other mechanisms that are not (strictly) dependent on the presence of specific CIs[18].

The concept of early metastatic spread in CRC[5-7] provides a framework for these observations. We suggest a model in which the early spread inherently involves tumor cells at early cytogenetic developmental stages, setting the foundation for distinct metastatic trajectories. When these early-disseminating tumor cells encounter the oxygen-rich pulmonary microenvironment, they appear capable of establishing metastases without requiring extensive chromosomal alterations. The favorable conditions of the lung environment could enable clonal expansion at a relatively early cytogenetic stage, resulting in lung metastases that can maintain this early cytogenetic profile. Although this oxygen-rich environment initially seems to offer conditions for metastatic growth, the notable enrichment of *KRAS* mutations in lung metastases[28] points to a more nuanced process. Specifically, *KRAS* mutations could thereby reduce the selective pressure for additional chromosomal imbalances. This stands in contrast to liver and brain metastases, where microenvironmental conditions might necessitate additional genomic alterations for successful colonization, in line with the more complex cytogenetic profiles. While our findings point toward minimal requirements for lung colonization, they do not exclude the possibility that metastatic cells continue to evolve and acquire additional chromosomal alterations within the lung microenvironment, nor do they preclude the successful establishment of lung metastases by tumor cells with more complex cytogenetic profiles. Indeed, our cohort and the MetTropism cohort[28] includes lung metastases displaying such advanced cytogenetic profiles.

Overall, the exploration of organotropic colonization signatures in metastatic CRC reveals a complex interplay of genetic factors that influence tumor behavior. Whereas the identified organotropic CIs suggest a predisposition for growth at specific metastatic sites, they do not provide a distinction between CRCs that metastasize and those that remain localized. This observation aligns with a growing body of evidence indicating that the genomic profiles of primary CRCs, regardless of metastatic spread, exhibit remarkable similarities in both mutation and CNV patterns[28,99]. Notably, only a limited subset of alterations—such as *SMAD4* mutations and *MYC* amplifications—has been significantly associated with metastatic CRC[28,99]. This observation underscores the need for further investigation of the mechanisms driving metastasis beyond invasion depths.

One potential approach for distinguishing metastatic potential in CRCs lies in examining the timing and sequence of acquiring specific CIs, which may play a critical role in shaping the biological behavior of cancer cells, including their metastatic potential. This notion parallels models proposed in the study of myeloproliferative neoplasms, where the order of mutations influences disease characteristics[100]. Supporting this perspective, recent research focusing exclusively on primary tumors has identified chromosomal gains at 7p and 8q as early events in the development of metastatic CRC[24]. In contrast, these same alterations were observed as late cytogenetic events in primary CRCs lacking clinical evidence of synchronous or metachronous metastases[24]. This temporal distinction suggests that the early acquisition of chromosomal gains at these loci may favor metastatic dissemination and could serve as biomarker for assessing metastatic potential in CRC. In light of these findings, exploring how early genetic changes might confer a selective advantage to tumor cells during the metastatic cascade represents a direction for future research.

In conclusion, our results support a model in which a site-specific pattern of cytogenetic events and the order in which particular CIs are acquired contribute to the potential of CRC to colonize particular secondary sites. Involvement of genes linked to signaling, chromatin/transcription, EMT, DNA repair and cell cycle may assist cancer cells to initiate and maintain a cascade that ultimately favors their survival in potentially challenging environments at the secondary site. The observed patterns of co-occurring genomic alterations and gene expression changes provide insights into the molecular mechanisms underlying CRC progression and metastasis, potentially offering approaches for targeted therapies and prognostic markers that predict the metastatic potential of CRC. Thus, the differential CIs described here add a cytogenetic layer to Stephen Paget's seed and soil hypothesis[13] and link cytogenetic events to the phenomenon of metastatic organotropism.

## Methods
### Cohorts
The study was approved by the local ethics committees (University Medical Center Göttingen, Ludwig Maximilian University of Munich) and included data published previously[60,63]. For the original data, the ethics committee of the University Medical Center Göttingen waived the requirement for informed consent. Data collection from the MSK MetTropism[28] and TCGA (PanCancer Atlas)[65,66] cohorts were collected by their respective research teams and made publicly available. The original studies reported that data collection was approved by their respective ethics committees, with informed consent obtained from patients. For the CRCTropism cohort, data comprised CI information, supplemented with the patient's sex and age at diagnosis as well as metastasis site. The sex was determined through self-reporting, combined with results from cytogenetic analyses. CIs in primary CRCs and metastases were determined by comparative genomic hybridization (CGH)[101]. Signal profiles were assigned for each chromosome by hierarchical cluster analysis. Threshold criteria were adjusted for the signal-to-background ratio to account for differences including tumor purity. We excluded the chromosomal regions 1p32pter, 13p, 14p, 15p, 21p, 22p, telomeres, and constitutive heterochromatic regions at 1q, 9q, 16q, and Yq from the analysis due to insufficient reliability as described previously[101], as well as individual chromosomal regions that did not pass quality criteria. Chromosomal arm aneuploidy was defined as chromosomal gain or loss that apparently encompassed the entire chromosomal arm[29]. Short arms of chromosomes 13, 14, 15, 21, and 22 were not considered for the aneuploidy count. Except where otherwise stated, aneuploidies and focal CIs were combined in the analyses. For statistical analysis, only one metastasis per patient was selected. In cases where multiple metastatic lesions were analyzed, we prioritized the selection based on their relative frequency within our dataset. Specifically, brain metastases were given the highest priority

due to their lower occurrence in the cohort, followed by lung metastases, and subsequently liver metastases. This prioritization ensured that our analysis reflected the characteristics of less common metastatic sites. When multiple metastases were present in the same location, we selected the one with the lowest number of CIs for analysis. This selection was based on the hypothesis that the metastasis with the fewest CIs might represent a state closer to the minimum genomic alterations potentially associated with metastatic spread.

The NGS-based MSK MetTropism cohort[28] was selected as independent cohort. Identified single/multi-nucleotide variants and indels, segmented copy number alterations (with subtraction from matched normals), the TMB, FGA, MSI score and clinical data were retrieved from cBioPortal[102,103]. The resource OncoKB[104] was used to classify mutations according to oncogenic potential. Only mutations listed as 'oncogenic' or 'likely oncogenic' were considered. Aneuploidies were determined by ASCETS[105] version 1.0 using default parameters. Gains and losses were analyzed using GISTIC 2.0[106] with a threshold for copy number amplifications and deletions of 0.3 and a $q$-value threshold of 0.05, conducted using GenePattern[107]. Only genes included in the MSK gene panel were considered in the analyses on the gene level. In addition, we obtained genomic and transcriptomic data from the TCGA project (PanCancer Atlas)[65,66]. The data, encompassing cancer subtype data, somatic mutations, CNVs, aneuploidies, transcriptomic profiles derived from RNA sequencing, mRNA expression z-scores relative to diploid samples, methylation data, MSIsensor[108] scores, as well as Winter hypoxia scores were accessed through the Genomic Data Commons Data (GCD) Portal[109], cBioPortal[102,103] as well as from a previous study[110]. CRCs classified as *POLE* subtype or unknown subtype were excluded.

## Functional proteomics data

RPPA data[68] on TCGA cancer samples were downloaded from the GDC data portal. Protein data included in the analysis were required to have successfully passed quality control (validation status 'valid').

## Co-occurrence modeling

The R[111] package cooccur[112] (version 1.3) was used to derive a probabilistic model of co-occurring CIs in CRC metastases. $P$-values obtained from the analyses were adjusted for multiple testing using the Benjamini-Hochberg method[113]. Co-occurrences are represented using a color scale: significant positive associations are indicated with shades of red, while significant negative associations are depicted in shades of blue. Associations that were not statistically significant and combinations that were unavailable are left blank.

## Organotropic mapping

Data were visualized in a ternary plot using the R package ggtern[114] built 3.5.0. A minimum occurrence threshold of three times was established for CI events to be included in the analysis. The relative frequency of each event observed in the cohort was calculated for the given metastasis site, and used to position the data point for each chromosomal arm in the organotropic map. The relative frequencies of each chromosomal aberration in the cohort were used to set the diameter of the respective bubble. To assess the statistical significance, two-sided Fisher exact tests, followed by $p$-value correction for multiple testing using the Benjamini-Hochberg method[113] were computed. Statistically significant chromosomal aberrations were defined as organotropic aberrations.

## Transcriptome and pathway analysis

For identification of differentially expressed genes, transcriptomic data were processed using edgeR[115] package version 4.0.3. For chromosome annotation, the package Organism.dplyr version 1.30.1 was used. Differential gene expression analysis was analyzed using two-sided Wilcoxon rank-sum tests with Benjamini-Hochberg adjustment[116] to compare the following CRC groups:

(I) *KRAS* mutation and amplification;
(II) *KRAS* mutation;
(III) *KRAS* wildtype;
(IV) *KRAS* deletion (but no oncogenic mutation);
(V) *MLH1* del: *MLH1* deletion, but no pathogenic mutation in *MLH1, MSH2, MSH6, PMS2*, no deletion of *MSH2, MSH6, PMS2*, no *MLH1* methylation defined as beta value < 0.3[117], no low expression of *MSH2, MSH6*, and *PMS2* defined as mRNA expression z-scores ≥ mean − one standard deviation (SD);
(VI) *MLH1* control: no pathogenic mutation and no deletion in the genes *MLH1, MSH2, MSH6*, and *PMS2*, no methylation defined as beta value < 0.3[117] and no low expression defined as mRNA expression z-scores ≥ mean − one SD, MSIsensor score < 3.5, not classified as MSI in cBioPortal;
(VII) *BRCA1* del: *BRCA1* deletion, but no pathogenic mutation in *BRCA1, BRCA2, PALB2, BARD1*, no deletion of *BRCA2, PALB2, BARD1*, no low expression of *BRCA2, PALB2, BARD1* defined as mRNA expression z-scores ≥ mean − one SD, MSIsensor score < 3.5, no *MLH1* methylation defined as beta value < 0.3[117];
(VIII) *BRCA1* control: in the genes *BRCA1, BRCA2, PALB2, RAD51C, BARD1* no pathogenic mutation and no deletion, no low expression defined as mRNA expression z-scores ≥ mean − one SD, MSIsensor score <3.5, no *MLH1* methylation defined as beta value < 0.3[117].

Differential gene expression data were visualized using volcano plots, with data points colored according to their respective chromosomal arms. Differentially expressed genes were defined using the following cut-offs for the log2(fold change) and statistical significance: 1.25-fold upregulation or downregulation and $p_{adjust} < 0.05$. For enrichment analyses, the following packages and parameters were used:

(I) Enrichr[118]: background gene list used, gene set definitions: GO[119,120], and MSigDB Hallmark[121,122];
(II) GSEA version 20.4.0[123], conducted in GenePattern[107]: permutation type: phenotype, scoring scheme: weighted, metric for ranking genes: Signal2Noise, normalization mode: meandiv, randomization mode: no_balance, gene set definitions: MSigDB[121,122];
(III) ssGSEA version 10.1.0[124], conducted in GenePattern[107]: weighting exponent 0.75, gene set definitions: MSigDB[121,122];
(IV) g:profiler version e111_eg58_p18_f463989d[125]: organism: *homo sapiens*, gene set definitions: GO[119,120], KEGG[126], and Reactome[127].

## Oncogenetic tree models

Oncogenetic tree models derived by maximum likelihood estimation were computed using the R[111] package oncomodel version 1.0[128]. The most common CIs found in the respective CRC metastasis type were considered for tree modeling. Chromosomal aberrations were grouped in nine classes according to their distance from the root of the oncogenetic tree (equal distances).

## Phylogenetic tree reconstruction using multi-region analyses

For patients with available CI data from multiple samples (primary CRC and at least one metastasis), CI events were encoded as discrete data. These data were then analyzed to infer phylogenetic relationships using the maximum parsimony method, as implemented in the software suite PHYLIP[129] version 3.697. The resultant phylogenetic trees were visualized using FigTree (version 1.4.4; http://tree.bio.ed.ac.uk/software/figtree/). CIs were color-encoded according to their classification within the oncogenetic trees.

## Data modeling

To investigate the relationship between *KRAS* copy number and MAP2K1/2 phosphorylation in *KRAS*-mutated CRCs, we employed a multiple linear regression model. The response variable in our model was the level of phosphorylated MAP2K1/2, which serves as a marker for MAP2K1/2 activation. Our primary predictor of interest was the copy number of *KRAS* (*KRAS*_CNV). Additionally, we included two other predictors: the total protein level of MAP2K1 (pMAP2K1) and the level of phosphorylated BRAF at serine 445 (pBRAF_S445), which serves as an indicator of BRAF. For linear regression modeling, the lm function implemented in the R package stats (built 4.3.1) was used. The significance of the *KRAS*_CNV effect was assessed using a likelihood ratio test comparing the full model to a reduced model without the *KRAS*_CNV term. We checked for multicollinearity among predictors using VIFs calculated with the car package (version 3.1.2). Variable importance was assessed using the vip package (version 0.4.1). Model diagnostics, including residual plots and tests for normality and homoscedasticity, were performed using base R functions and the lmtest package (version 0.9.40). To assess the evolutionary trajectories, we calculated CI scores, which correspond to the average of the CI event categories derived from the phylogenetic trees. Linear mixed-effect modeling was conducted using the package lme4[130] (version 1.1.35.5), with tumor evolutionary step as fixed effect and tumors as random effect to analyze CI scores. Post-hoc analyses with Benjamini-Hochberg adjustment[113] were performed following the modeling.

## Further statistics, reproducibility and data visualization

Statistical analyses were performed using two-sided tests. The Wilcoxon rank-sum test with continuity correction, Kruskal-Wallis tests with Dunn post-hoc tests and Fisher exact test for contingency tables were used, as indicated. To address multiple testing, *p*-values were corrected using the Benjamini-Hochberg procedure[113]. The significance level was set at $p < 0.05$ (when applicable, after correction for multiple testing). The statistical environment $R$[111] (version 4.3.3) was used for statistical analysis and data visualization using ggplot2 version 3.5.1. No statistical method was used to predetermine sample size. No data that passed quality control were excluded from the analyses. The experiments were not randomized. The Investigators were not blinded to allocation during experiments and outcome assessment.

## Reporting summary

Further information on research design is available in the Nature Portfolio Reporting Summary linked to this article.

## Data availability

The previously published CGH data[60,63] comprise chromosomal imbalance profiles indicating net clonal changes, characterized by regions of chromosomal losses, gains, and amplifications at the chromosomal band level, accompanied by basic demographic information including sex and age. The MSK MetTropism publicly available data[28] used in this study are available in the cBioPortal database (https://www.cbioportal.org) under accession code MSK MetTropsim. The TCGA publicly available data[65,66,68] used in this study are available at the GDC Portal (https://portal.gdc.cancer.gov) under accession codes TCGA-COAD and TCGA-READ and cBioPortal under accession code Colorectal Adenocarcinoma (TCGA, PanCancer Atlas). The remaining data are available within the Article, Supplementary Information or Source Data file. Source data are provided with this paper.

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

## Acknowledgements

We thank Christina Enders and Cynthia Bunker for technical support, and thank the researchers at the Memorial Sloan Kettering Cancer Center, New York, the TCGA network and the Functional Proteomics Reverse Phase Protein Array Core at the University of Texas MD Anderson Cancer Center for making their data available. This study was supported by the guest professorship program of the University of Augsburg and the professorship of the Faculty of Medicine at the University of Augsburg to MMG.

## Author contributions

Conceptualization: M.M.G., L.F., B.S. Methodology: M.M.G, B.S. Formal analysis: M.M.G., B.S. Investigation: M.M.G., B.S. Resources: M.M.G., B.G., A.G., B.C.D., C.S., L.F., T.L., B.S. Data curation: B.G., J.S.G., L.F. Writing – Original Draft: M.M.G. Writing – Review & Editing: all authors. Visualization: M.M.G., B.S. Supervision: M.M.G., L.F., B.S. Project administration: M.M.G. Funding acquisition: M.M.G.

## Funding

## Competing interests

The authors are not aware of any conflict of interest related to this study.
