## [Transparent Peer Review file · Nature Communications]

Cytogenetic Signatures Favoring Metastatic Organotropism in Colorectal Cancer

Corresponding Author: Professor Mariola Monika Golas

Version 0:

Reviewer comments:

Reviewer #1

(Remarks to the Author)

The authors analyzed the CRCTropism cohort for investigating metastatic organotropism, which is a key question in the field. Major findings include (1) brain metastasis has the highest number of CIs; (2) site-specific patterns of CI co-occurrence were observed; (3) evolutionary analyses revealed CI might happen early and contribute to site-specific patterns of tumor metastasis. I have several concerns and/or suggestions:

1. Most of the current observations have been reported before. The authors should adjust the manuscript structure to further highlight the novel findings;
2. Tumor purity has not been considered for the copy number analysis;
3. Polygenetic tree of each individual patient should be included for patients with longitudinal samples;
4. The accuracy of copy number analysis is unknown. It's better to benchmark the results by comparison analysis between the current study and well-established CRC cohort, such as TCGA;
5. Figure 1 should also include some Gistic plots to be more informative;
6. What are the potential molecular mechanism of the observed patterns of CI organotropism?

Reviewer #2

(Remarks to the Author)

The submitted manuscript measures copy number variations (CNV) or chromosome imbalances in two different cohorts with two different methods (CGH and sequencing) on metastatic CRC. There is a plethora of data and analysis, and the basic conclusion is that some CNVs are more common (or "organotropic") in either liver, lung or brain metastatic sites. Overall the findings are interesting. Several comments:

- 1) The number of lesions is a bit hard to follow. Page 5 notes 3,507 lesions but the Results (page 8) has 314 tumors.
- 2) Critical to the analysis is sample purity (% tumor in the sample) because CNVs are harder to detect if there are too many normal cells in the sample. This technical issue could lead to greater average CNVs in the brain and liver (pg 8), which tend to form solid masses relatively free of stromal cells. Is there a way to estimate and correct for sample purity? Some comments on this issue seem merited.
- 3) The assignment of genes to the CI (page 13) is based on the MSK gene panel, which is biased towards driver mutations. The paper indicates that a gene is "altered" if it is included in a CI. However, there are multiple genes typically included on any CNV or CI, and it is difficult to state that these genes and their pathways (Fig 4d) are specifically activated or enriched when there is a CI or CNV. Moreover, because the MSK panel is enriched for driver mutations, any pathway enrichment analysis will be enriched for critical cellular pathways. Typically, pathway analysis survey much larger sets of genes to infer

enrichment. The paper could either delete the gene analysis or much better substantiate how a gene on a CI (whether a tumor suppressor or oncogene) inevitably becomes a driver alteration.

4) Ordering of events (page 15): The paper takes the approach that alteration frequency determines “when” it occurs, akin to the Vogelgram. Although often done, this approach can be misleading because many individual tumors “violate” the “typical” pattern. For example, loss of 5q (APC!) is considered a “late” event (page 16, line 359), but in Fig 6c, tumors C and G have del5q at their roots. This section should reflect or discuss that the typical proposed pattern is often not seen in individual tumors.

5) Discussion: A general comment is that because nonmetastatic CRCs were not directly compared, it is difficult to infer that the organotropic CI are driving the metastatic process. The paper does address that when a metastasis occurs, there is enrichment of certain CI at certain sites. But there are also nonmetastatic CRCs with the same CIs. This potential problem is somewhat addressed in the Discussion (page 18, line 416), but a bit more discussion might be merited.

6) Discussion: The linkage of a CI to the activation of a specific MSK gene is not well justified because between most CI changed do not often “activate” the genes on the CI. There are so many CNVs in CRCs, and the Discussion is unfocused relative to the actual data in the manuscript. Need to better justify why the authors feel a CI will inevitably activate their annotated genes.

Version 1:

Reviewer comments:

Reviewer #1

(Remarks to the Author)

The authors have addressed my concerns.

Reviewer #2

(Remarks to the Author)

The revise manuscript addresses my major concerns. In particular, the mechanism on how CNVs might alter tumorigenesis is much better addressed (versus the mere activation of a gene on a CNV), with a nice exploration of KRAS and 12p increases in brain metasases. The manuscript is shorter, more focused and easier to read, and the progression to metastasis is better outlined. Metastasis mechanisms are still largely unexplained and the large amount of data with multiple metastatic sites would be of interest to many other investigators.

Minor point

1) Page 7 line 128--- would be nice to just state the fold changes as well as so a reader does not have to go searching.

Point-by-point reply to the Reviewers' comments:

Reviewer #1:

The authors analyzed the CRCTropism cohort for investigating metastatic organotropism, which is a key question in the field. Major findings include (1) brain metastasis has the highest number of CIs; (2) site-specific patterns of CI co-occurrence were observed; (3) evolutionary analyses revealed CI might happen early and contribute to site-specific patterns of tumor metastasis. I have several concerns and/or suggestions:

1. Most of the current observations have been reported before. The authors should adjust the manuscript structure to further highlight the novel findings;

The entire manuscript has been revised to include new results that suggest mechanisms for metastatic organotropism while condensing the text to enhance readability and avoid excessive length. For further details, please refer to our detailed responses below. Additionally, we have revised our statistical analyses to include only one case per patient where appropriate, ensuring the independence of the data in accordance with the requirements of statistical tests.

2. Tumor purity has not been considered for the copy number analysis;

We carefully considered tumor purity in our copy number analyses, recognizing its critical role in accurately interpreting genomic data. Specifically, the thresholds for defining copy number gains and losses were tailored to each sample based on the signal-to-background characteristics, which varies based on technical factors, especially tumor purity. Furthermore, we employed hierarchical cluster analysis to align signal profiles on an individual basis for each chromosome, ensuring a robust assessment of the genomic landscape. To provide greater clarity on these methodologies, we have revised the methods section of the manuscript.

Please see also reply to comment 2 of reviewer 2.

3. Polygenetic tree of each individual patient should be included for patients with longitudinal samples.

The phylogenetic trees for patients with multiple primary tumor-metastasis samples, which were not included in Figure 7 of the main manuscript, are now presented in the multi-page Supplementary Figure S6. This addition provides a comprehensive view of the evolutionary relationships among these samples, further enriching our analysis.

4. The accuracy of copy number analysis is unknown. It's better to benchmark the results by comparison analysis between the current study and well-established CRC cohort, such as TCGA;

To validate our findings and ensure the robustness of our results, we conducted a benchmarking analysis by comparing our data with the TCGA dataset. Specifically, we examined the frequencies of chromosomal imbalances in our cohort relative to those in the TCGA dataset. Given that the TCGA-CRC dataset comprises exclusively primary cancer samples, we focused our benchmarking analysis on primary CRCs from our cohort. It is important to note that differences in chromosomal imbalance patterns would be anticipated when comparing primary CRC samples to metastatic sites, particularly regarding differences in the organotropic chromosomal imbalances.

Our comparative analysis was performed separately for primary CRCs that metastasized to the liver, lung, and brain. Overall, the aneuploidy profiles exhibited a remarkably similar pattern, with no statistically significant differences observed between primary CRCs with liver and brain metastases *versus* the TCGA dataset (Figure R1). However, for primary CRCs that disseminated to the lung, we detected significant differences in specific chromosomal regions, including a lower frequency of loss at 18p, a higher frequency of loss at 20p, and a reduced frequency of gain at 20p. Collectively, these findings suggest that our study aligns well with the TCGA dataset while highlighting specific differences, which we suggest to reflect the underlying biological behavior and metastatic propensity of the tumors, consistent with the chromosomal imbalance profiles observed in their respective metastases. The results of this benchmarking analysis are also detailed in Supplementary Figure S2 of the Supplementary Information.

Figure R1: Chromosomal imbalance profile of primary CRC.

Additionally, we performed an individual analysis of chromosomal profiles. For chromosome 8, for example, well-established genetic patterns include loss at 8p and gain at 8q, with chromosomal breakpoints enriched around chromosomal band 8p11/8p12 (compare reference ¹). In addition, an enrichment of amplifications is expected at the *MYC* locus, which may exhibit high-level amplifications in various cancers including CRC. This expected pattern of chromosomal imbalances was confirmed in our data, including a maximum around the *MYC* gene (Figure R2). These data are also included as Supplementary Figure S4.

Figure R2: Chromosomal imbalance profile of chromosome 8 in our CRCTropism cohort. The solid line depicts the mean signal ratio (\log_2) across the chromosome, with dashed lines representing one standard deviation above and below the mean. The horizontal dotted line indicates the baseline for no imbalance.

Breakpoints are predominantly observed in the proximal region of the chromosomal 8p arm, in line with previous data¹. The greyish area indicates the maximum around the *MYC* gene (arrowhead). For orientation, an ideogram of chromosome 8 is provided at the top.

5. Figure 1 should also include some Gistic plots to be more informative;

GISTIC plots are presented in Supplementary Figure S3 of the revised manuscript. In addition to the focal chromosomal aberrations typically observed in CRCs, the GISTIC analysis of the CRCTropism cohort revealed several additional aberrations that have not been commonly reported in primary CRC. Amongst others, the analysis identified a significant gain in the 5q35.2 chromosomal region. This region is shown in our study to represent a brain-enriched organotropic chromosomal imbalance and is rarely observed in other metastatic sites or in primary CRC.

6. What are the potential molecular mechanism of the observed patterns of CI organotropism?

To address molecular mechanisms, we performed a set of additional analyses utilizing RNA-seq data from The Cancer Genome Atlas (TCGA), combined with CNV, mutation and proteomic data. In this reply, we summarize some of the key findings, with a full description provided in Figures 4 and 5 of the revised manuscript, along with detailed presentation in the revised results and discussion sections.

In particular, we studied CRCs with oncogenic *KRAS* mutations, comparing those with *KRAS* amplification plus oncogenic *KRAS* mutation to those harboring an oncogenic *KRAS* mutation alone. The additional amplification of *KRAS* resulted in significantly higher *KRAS* expression, and the *KRAS_CNV* was a significant predictor of phosphorylated MAP2K1/2, downstream effectors in the RAS pathway. The enhanced *KRAS* expression was reflected in changes to the overall gene expression profile, including the upregulation of gene sets positively linked to oncogenic *KRAS* expression, such as those associated with glycolysis. This metabolic rewiring seems particularly relevant for brain metastatic cells, which are believed to benefit from enhanced glycolytic activity. Such adaptations provide several advantages, including increased biosynthetic capacity of biomolecules to support cancer growth, adaptability to the brain microenvironment, and an invasive, acid-resistant phenotype — all of which suggested to facilitate colonization of the brain niche².

Moreover, we observed that in cells with both *KRAS* mutations and amplifications, there was also upregulation of other genes located on chromosome 12 (where *KRAS* is encoded at 12p12.1), including *CDK4* and *MDM2*. This observation aligns with our findings that the MDM2/cell cycle pathway is upregulated. The co-amplification of *KRAS* and other genes on chromosome 12 may further drive tumor progression and metastasis.

In addition to these findings related to *KRAS* alterations, we explored other genetic changes implicated in metastatic organotropism, specifically focusing on CRCs with deletions of DNA repair genes *MLH1* and *BRCA1*— both of which are organotropically enriched in brain metastases. In CRCs with *MLH1* deletion, we identified significant patterns of gene expression changes including members of the *HOX* gene family. The activation of *HOX* genes in these cancers may contribute to their aggressive behavior by influencing critical biological processes such as cell proliferation, differentiation, and epithelial-mesenchymal transition (EMT). Overall, our findings highlight a complex molecular landscape associated with CRCs exhibiting concurrent *KRAS* amplification and mutation as well as deletions of *MLH1* and *BRCA1*.

Reviewer #2:

The submitted manuscript measures copy number variations (CNV) or chromosome imbalances in two different cohorts with two different methods (CGH and sequencing) on metastatic CRC. There is a plethora of data and analysis, and the basic conclusion is that some CNVs are more common (or “organotropic”) in either liver, lung or brain metastatic sites. Overall the findings are interesting. Several comments:

1) The number of lesions is a bit hard to follow. Page 5 notes 3,507 lesions but the Results (page 8) has 314 tumors.

We have revised the paragraphs to enhance clarity and avoid any potential misunderstanding. Since our study involves distinct cohorts, we initially aggregated all lesions from each cohort in the first version of the manuscript. In the revised manuscript, we now discuss each cohort separately, providing a clearer

understanding of the data. Additionally, we have included the number of lesions for each cohort in Supplementary Tables S1, S4, and S7 to facilitate easier reference and comparison:

CRCTropism cohort	314 lesions (metastases: 117 liver, 78 lung, 39 brain)
MSK MetTropism cohort	3,548 lesions (metastases: 624 liver, 146 lung, 22 brain)
TCGA	449 (no metastases)

2) Critical to the analysis is sample purity (% tumor in the sample) because CNVs are harder to detect if there are too many normal cells in the sample. This technical issue could lead to greater average CNVs in the brain and liver (pg 8), which tend to form solid masses relatively free of stromal cells. Is there a way to estimate and correct for sample purity? Some comments on this issue seem merited.

We appreciate the reviewer's comment regarding the potential impact of low tumor sample purity on the identification of genetic alterations. In this study, we made a concerted effort to select tissue specimens with the highest possible tumor cell content, as assessed by experienced pathologists. The sample purity typically ranged from 50% to 75%. Furthermore, we adjusted our analytical thresholds based on signal-to-background characteristics, which depend on sample purity, amongst others. Please also refer to our answer to comments 2 and 4 from reviewer 1.

Additionally, we wish to highlight that not only did the lung metastases exhibit fewer chromosomal aberrations than liver and brain metastases, but the corresponding primary CRCs also displayed a similar trend. Specifically, both showed a lower frequency of loss at 18p and a reduced frequency of gain at 20p—genetic alterations identified in this study as associated with brain- and liver-specific organotropism (see also Figure R1).

3) The assignment of genes to the CI (page 13) is based on the MSK gene panel, which is biased towards driver mutations. The paper indicates that a gene is "altered" if it is included in a CI. However, there are multiple genes typically included on any CNV or CI, and it is difficult to state that these genes and their pathways (Fig 4d) are specifically activated or enriched when there is a CI or CNV. Moreover, because the MSK panel is enriched for driver mutations, any pathway enrichment analysis will be enriched for critical cellular pathways. Typically, pathway analysis survey much larger sets of genes to infer enrichment. The

paper could either delete the gene analysis or much better substantiate how a gene on a CI (whether a tumor suppressor or oncogene) inevitably becomes a driver alteration.

We appreciate the reviewer's comments regarding the enrichment of oncogenes and tumor suppressor genes in the MSK cancer gene panel. In response to this comment, we have replaced the pathway analysis with a comprehensive examination of transcriptomic data to strengthen our findings. Specifically, we conducted an extensive analysis using the TCGA dataset, which enabled us to perform an integrative assessment of genomic alterations and their implications for CRC.

Our analysis revealed significant rewiring of CRCs with *KRAS* mutation and amplification, also in comparison to CRCs with *KRAS* mutation only. This rewiring was linked to the upregulation of glycolytic genes, which may confer metastatic cells with energetic and biosynthetic advantages, as well as invasive characteristics that could facilitate colonization of the brain microenvironment. For further details, please refer to our response to point 6 from reviewer 1. A full description of these results can be found in Figures 4 and 5 of the revised manuscript, along with the corresponding sections in the results and discussion.

4) Ordering of events (page 15): The paper takes the approach that alteration frequency determines "when" it occurs, akin to the Vogelgram. Although often done, this approach can be misleading because many individual tumors "violate" the "typical" pattern. For example, loss of 5q (APC!) is considered a "late" event (page 16, line 359), but in Fig 6c, tumors C and G have del5q at their roots. This section should reflect or discuss that the typical proposed pattern is often not seen in individual tumors.

We fully agree with the reviewer that the order of genetic events in CRC is inherently individual and can vary significantly among tumors. As highlighted in the landmark manuscript by Fearon and Vogelstein³, the specific sequence of genetic alterations may not be as critical as the overall accumulation of these changes over time. This perspective aligns with the evolutionary trajectories observed in the CRCs presented in our study, as illustrated in Figure 7 and Supplementary Figure S6.

Overall, our data indicate that certain chromosomal aberrations tend to be preferentially found early or late in the evolutionary trajectory of CRC. However, this does not preclude the possibility that individual lesions may have acquired genetic alterations concurrently, subsequently, or prior to these

identified aberrations. Such genetic changes may create a permissive environment that facilitates the acquisition of further driver alterations and promotes the clonal expansion of transformed cells.

To substantiate our findings, we employed linear mixed-effects modeling to analyze our data. This approach allows for individual trajectories while supporting events classified as early, intermediate, or late are preferentially observed at their respective stages within the oncogenetic trees. The results of these new analyses are presented in Supplementary Figure S7, along with corresponding sections in the results. For convenience, we have also included relevant data in this response as Figure R3. Additionally, we have revised our discussion to emphasize this point more clearly and to acknowledge the variability in the order of genetic events during CRC evolution.

Figure R3: Classification of CI events based on patient-specific phylogenetic trees. Box-and-whisker plots display the distribution of CI scores for early, intermediate, and late temporal categories, as derived from individual patients' phylogenetic trees. The CI event class is derived from the oncogenetic trees (see Figure 6 in the main manuscript). Each plot includes a dot indicating the mean score for that category. Linear mixed-effect modeling was employed to assess statistical significance between temporal categories, followed by post-hoc testing with Benjamini-Hochberg correction. Significant differences are indicated by asterisks (**, $p < 0.01$; ***, $p < 0.001$).

5) Discussion: A general comment is that because nonmetastatic CRCs were not directly compared, it is difficult to infer that the organotropic CI are driving the metastatic process. The paper does address that when a metastasis occurs, there is enrichment of certain CI at certain sites. But there are also nonmetastatic CRCs with the same CIs. This potential problem is somewhat addressed in the Discussion (page 18, line 416), but a bit more discussion might be merited.

As suggested, we have added the following paragraph to the discussion section: “Overall, the exploration of organotropic colonization signatures in metastatic CRC reveals a complex interplay of genetic factors that influence tumor behavior. Whereas the identified organotropic CIs suggest a predisposition for growth at specific metastatic sites, they do not provide a distinction between CRCs that metastasize and those that remain localized. This observation aligns with a growing body of evidence indicating that the genomic profiles of primary CRCs, regardless of metastatic spread, exhibit remarkable similarities in both mutation and CNV patterns^{4,5}. Notably, only a limited subset of alterations—such as *SMAD4* mutations and *MYC* amplifications—has been significantly associated with metastatic CRC^{4,5}. This observation underscores the need for further investigation into the mechanisms driving metastasis beyond invasion depths.

One potential approach for distinguishing metastatic potential in CRCs lies in examining the timing and sequence of acquiring specific CIs, which may play a critical role in shaping the biological behavior of cancer cells, including their metastatic potential. This notion parallels models proposed in the study of myeloproliferative neoplasms, where the order of mutations influences disease characteristics⁶. Supporting this perspective, recent research focusing exclusively on primary tumors has identified chromosomal gains at 7p and 8q as early events in the development of metastatic CRC⁷. In contrast, these same alterations were observed as late cytogenetic events in primary CRCs lacking clinical evidence of synchronous or metachronous metastasis⁷. This temporal distinction suggests that the early acquisition of chromosomal gains at these loci may favor metastatic dissemination and could serve as biomarker for assessing metastatic potential in CRC. In light of these findings, exploring how early genetic changes might confer a selective advantage to tumor cells during the metastatic cascade represents a direction for future research.”

6) Discussion: The linkage of a CI to the activation of a specific MSK gene is not well justified because between most CI changed do not often “activate” the genes on the CI. There are so many CNVs in CRCs,

and the Discussion is unfocused relative to the actual data in the manuscript. Need to better justify why the authors feel a CI will inevitably activate their annotated genes.

As outlined in our responses to point 6 of reviewer 1 and point 3 of reviewer 2, we have conducted additional analyses. Based on these results, we have revised the discussion section accordingly.

References

1. Palin, K. *et al.* Contribution of allelic imbalance to colorectal cancer. *Nat Commun* **9**, 3664 (2018).
2. Gatenby, R. A. & Gillies, R. J. Why do cancers have high aerobic glycolysis? *Nat Rev Cancer* **4**, 891-899 (2004).
3. Fearon, E. R. & Vogelstein, B. A genetic model for colorectal tumorigenesis. *Cell* **61**, 759-767 (1990).
4. Li, C. *et al.* Integrated Omics of Metastatic Colorectal Cancer. *Cancer Cell* **38**, 734-747 e739 (2020).
5. Nguyen, B. *et al.* Genomic characterization of metastatic patterns from prospective clinical sequencing of 25,000 patients. *Cell* **185**, 563-575 e511 (2022).
6. Ortmann, C. A. *et al.* Effect of mutation order on myeloproliferative neoplasms. *N Engl J Med* **372**, 601-612 (2015).
7. Golas, M. M. *et al.* Evolutionary patterns of chromosomal and microsatellite instability in proximal and distal colorectal cancer. *Colorectal Dis* **24**, 157-176 (2022).

Point-by-point response to the reviewer

Reviewer #1 (Remarks to the Author):

The authors have addressed my concerns.

Reviewer #2 (Remarks to the Author):

The revised manuscript addresses my major concerns. In particular, the mechanism on how CNVs might alter tumorigenesis is much better addressed (versus the mere activation of a gene on a CNV), with a nice exploration of KRAS and 12p increases in brain metastases. The manuscript is shorter, more focused and easier to read, and the progression to metastasis is better outlined. Metastasis mechanisms are still largely unexplained and the large amount of data with multiple metastatic sites would be of interest to many other investigators.

Minor point

1) Page 7 line 128--- would be nice to just state the fold changes as well as so a reader does not have to go searching.

We have incorporated the requested information into the revised manuscript and, for consistency, added these data to a second paragraph as well.